# AH-UGC: A̲daptive and H̲eterogeneous-U̲niversal G̲raph C̲oarsening

## Abstract

**Graph Coarsening (GC)** is a prominent graph reduction technique that compresses large graphs to enable efficient learning on graphs. However, existing GC methods generate only one coarsened graph per run and must recompute from scratch for each new coarsening ratio, resulting in unnecessary overhead. Moreover, most prior approaches are tailored to *homogeneous* graphs and fail to accommodate the semantic constraints of *heterogeneous* graphs, which comprise multiple node and edge types. To overcome these limitations, we introduce a novel framework that combines Locality-Sensitive Hashing (LSH) with Consistent Hashing (CH) to enable *adaptive graph coarsening*. Leveraging hashing techniques, our method is inherently fast and scalable. For heterogeneous graphs, we propose a *type-isolated coarsening* strategy that ensures semantic consistency by restricting merges to nodes of the same type. Our approach is the first unified framework to support both adaptive and heterogeneous coarsening. Extensive evaluations on 23 real-world datasets including homophilic, heterophilic, homogeneous, and heterogeneous graphs demonstrate that our method achieves superior scalability while preserving the structural and semantic integrity of the original graph. Our code is available here.

## 1 Introduction

Graphs are ubiquitous and have emerged as a fundamental data structure in numerous real-world applications Kataria et al. (2025); Fout et al. (2017); Wu et al. (2020). Broadly, graphs can be categorized into two types: (a) *Homogeneous graphs* Shchur et al. (2018); Wang et al. (2020), which consist of a single type of nodes and edges. For instance, in a homogeneous citation graph, all nodes represent papers, and all edges represent the "cite" relation between them; (b) *Heterogeneous graphs* Liu et al. (2023a); Yang et al. (2020); Lv et al. (2021), which involve multiple types of nodes and/or edges, enabling the modeling of richer and more realistic interactions. For example, in a recommendation system, a heterogeneous graph may contain nodes of different types, such as users, items, and categories, and edge types such as "(user, buys, item)", "(user, views, item)", and "(item, belongs-to, category)". Although many real-world datasets are inherently heterogeneous, early research in graph machine learning predominantly focused on homogeneous graphs due to their modeling simplicity, availability of standardized benchmarks, and theoretical tractability Dwivedi et al. (2023); Lim et al. (2021). However, the limitations of homogeneous representations in capturing rich semantic information have shifted attention toward heterogeneous graph modeling Yang et al. (2020); Zhang et al. (2019).

As real-world networks continue to grow rapidly in size and complexity, large-scale graphs have become increasingly common across various domains Kong et al. (2023); Zeng et al. (2019); Bhatia et al. (2016). This surge in scale poses significant computational and memory challenges for learning and inference tasks on such graphs. This underscores the growing importance of developing efficient and effective methodologies for processing large-scale graph data. To address the issue, an expanding line of research investigates graph reduction methods that compress structures without compromising essential properties. Most existing graph reduction techniques, including pooling Bianchi et al. (2020), sampling-based Dhillon et al. (2007), condensation Jin et al. (2021b), and coarsening-based methods Kumar et al. (2023); Kataria et al. (2024); Loukas (2019). Coarsening methods have demonstrated effectiveness in preserving structural and semantic information Loukas (2019); Kumar et al. (2023); Kataria et al. (2024), this study focuses on graph coarsening (GC) as the primary reduction strategy. Despite advancements in existing GC frameworks, two key challenges remain:

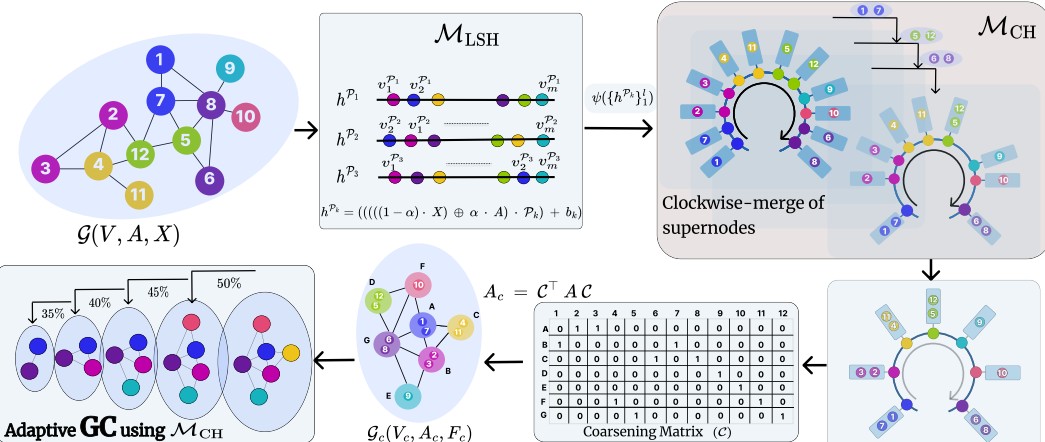

Figure 1: AH-UGC consists of three modules: (a) $\mathcal{M}_{\text{LSH}}$ constructs an augmented feature matrix by combining node features and structural context using a heterophily-aware factor $\alpha$, enabling support for both homophilic and heterophilic graphs. Inspired by UGC Kataria et al. (2024), we use LSH projections to compute node hash indices via $\psi(h^{(\mathcal{P}_k)^l}_1)$ (see Section 3); (b) $\mathcal{M}_{\text{CH}}$ applies consistent hashing to merge nodes clockwise based on a target coarsening ratio $r$, yielding the coarsening matrix $\mathcal{C}$; (c) the coarsened graph $\mathcal{G}_c$ is obtained via $A_c = \mathcal{C}^\top A \mathcal{C}$. The framework is inherently adaptive— i.e., once an intermediate coarsening is obtained, further reduction can be applied incrementally using $\mathcal{M}_{\text{CH}}$ and already calculated coarsening matrix $\mathcal{C}$, enabling efficient multi-resolution processing.

- **Lack of "Adaptive Reduction".** Many applications, such as interactive visualization and real-time recommendations, benefit from multi-resolution graph representations. These scenarios often require dynamically adjusting the coarsening ratio based on user interaction or task demands. However, most existing methods generate a single fixed-size coarsened graph and must recompute from scratch for each new ratio, incurring high overhead. This highlights the need for adaptive coarsening frameworks that enable efficient, progressive refinement without redundant computation.
- **Lack of "Heterogeneous Graph Coarsening" Framework.** Existing methods typically assume homogeneous node types, making them unsuitable for heterogeneous graphs with semantically distinct nodes. This can result in invalid supernodes for example, merging an *author* with a *paper* node in a citation graph thus violating type semantics. Moreover, node types often have different feature dimensions, which standard coarsening techniques are not designed to handle.

**Key Contribution.** To address the dual challenges of adaptive reduction and heterogeneous GC, we propose **AH-UGC**, a unified framework for Adaptive and Heterogeneous Universal Graph Coarsening. We integrate locality-sensitive hashing (LSH) Datar et al. (2004) with consistent hashing (CH) Karger et al. (1997). While LSH ensures that similar nodes are coarsened together based on their features and connectivity Kataria et al. (2023; 2024), CH—a technique originally developed for load balancing Chen et al. (2021), enables us to design a coarsening process that supports multi-level adaptive coarsening without reprocessing the full graph. To handle heterogeneous graphs, AH-UGC enforces *type-isolated coarsening*, wherein nodes are first grouped by their types, and coarsening is applied independently within each type group. This ensures that nodes and edges of incompatible types are never merged, preserving the semantic structure of the original heterogeneous graph. Additionally, AH-UGC is naturally suited for streaming or evolving graph settings, where new nodes and edges arrive over time. Our LSH- and CH-based method allows new nodes to be integrated into the existing coarsened structure with minimal recomputation. To summarize, **AH-UGC** is a general-purpose graph coarsening framework that supports *adaptive, streaming, expanding, heterophilic, and heterogeneous graphs*.

## 2 BACKGROUND

We first discuss the necessary background, problem formulation, and related work in this section. A more detailed description of all symbols, along with a comprehensive notation table, is provided in Appendix A.

**Definition 1 (Graph)** *A graph is represented as $\mathcal{G}(V, A, X)$, where $V = \{v_1, \ldots, v_N\}$ is the set of $N$ nodes, $A \in \mathbb{R}^{N \times N}$ is the adjacency matrix, and $X \in \mathbb{R}^{N \times \widetilde{d}}$ is the node feature matrix with each row $X_i \in \mathbb{R}^{\widetilde{d}}$ denoting the feature vector of node $v_i$. An edge between nodes $v_i$ and $v_j$ is indicated by $A_{ij} > 0$. Let $D \in \mathbb{R}^{N \times N}$ be the degree matrix with $D_{ii} = \sum_j A_{ij}$ then $L = D - A$ denotes the Laplacian matrix. $L \in S_L$, where $S_L = \left\{ L \in \mathbb{R}^{N \times N} \,\middle|\, L_{ij} = L_{ji} \leq 0 \text{ for } i \neq j; L_{ii} = -\sum_{j \neq i} L_{ij} \right\}$. For $i \neq j$, the matrices are related by $A_{ij} = -L_{ij}$, and $A_{ii} = 0$. Hence, the graph $\mathcal{G}(V, A, X)$ may equivalently be denoted $\mathcal{G}(L, X)$, and we use either form as contextually appropriate.*

**Definition 2 (Heterogeneous graph)** *A heterogeneous graph can be represented in two equivalent forms, with either representation utilized as required within the paper.*

- **Entity-based:** *A heterogeneous graph extends the standard graph structure by incorporating multiple types of nodes and/or edges. Formally, a heterogeneous graph is defined as $\mathcal{G}(V, E, \Phi, \Psi)$, where $\Phi : V \to \mathcal{T}_V$ and $\Psi : E \to \mathcal{T}_E$ are node-type and edge-type mapping functions, respectively (Lv et al., 2021). Here, $\mathcal{T}_V$ and $\mathcal{T}_E$ denote the sets of possible node types and edge types. When the total number of node types $|\mathcal{T}_V|$ and edge types $|\mathcal{T}_E|$ is equal to 1, the graph degenerates into a standard homogeneous graph (Definition 1).*
- **Type-based:** *Alternatively, a heterogeneous graph can be described as $\mathcal{G}(\{X_{(node\_type)}\}, \{A_{(edge\_type)}\}, \{y_{(target\_type)}\})$, where feature matrices $X$, adjacency matrices $A$, and target labels $y$ are grouped and indexed by their corresponding node, edge, and target types (Gao et al., 2024a).*

**Definition 3 (Graph Coarsening)** *Following Loukas (2019); Kataria et al. (2024); Kumar et al. (2023), The **G**raph **C**oarsening (GC) problem involves learning a coarsening matrix $\mathcal{C} \in \mathbb{R}^{N \times n}$, which linearly maps nodes from the original graph $\mathcal{G}$ to a reduced graph $\mathcal{G}_c$, i.e., $V \to \widetilde{V}$. This linear mapping should ensure that similar nodes in $\mathcal{G}$ are grouped into the same super-node in $\mathcal{G}_c$, such that the coarsened feature matrix is given by $\widetilde{X} = \mathcal{C}^\top X$. Each non-zero entry $\mathcal{C}_{ij}$ denotes the assignment of node $v_i$ to super-node $\widetilde{v}_j$. The matrix $\mathcal{C}$ must satisfy the following structural constraints:*

$$\mathcal{S} = \{\mathcal{C} \in \mathbb{R}^{N \times n}, \ \mathcal{C}_{ij} \in \{0, 1\}, \ \|\mathcal{C}_i\| = 1, \ \langle \mathcal{C}_i^\top, \mathcal{C}_j^\top \rangle = 0 \ \forall i \neq j, \ \langle \mathcal{C}_l^\top, \mathcal{C}_l^\top \rangle = d_{\widetilde{V}_l}, \ \|\mathcal{C}_i^\top\|_0 \geq 1\}$$

*where $d_{\widetilde{V}_l}$ means the number of nodes in the $l^{th}$-supernode. The condition $\langle \mathcal{C}_i^\top, \mathcal{C}_j^\top \rangle = 0$ ensures that each node of $\mathcal{G}$ is mapped to a unique super-node. The constraint $\|\mathcal{C}_i^\top\|_0 \geq 1$ requires that each super-node contains at least one node.*

### 2.1 PROBLEM FORMULATION AND RELATED WORK

We formalize the problem through two key objectives: Goal 1. Adaptive Coarsening and Goal 2. Graph Coarsening for Heterogeneous Graphs.

**Goal 1.** The objective is to compute multiple coarsened graphs $\{\mathcal{G}_c^{(r)}\}_{r=1}^R$ from input graph $\mathcal{G}(V, A, X)$, where each $\mathcal{G}_c^{(r)}$ corresponds to a target coarsening ratio $r \in (0, 1]$, without recomputing from scratch for each resolution. Formally, the goal is to construct a family of coarsening matrices $\{\mathcal{C}^{(r)} \in \mathbb{R}^{N \times n^{(r)}}\}$ such that

$$\widetilde{X}^{(r)} = (\mathcal{C}^{(r)})^\top X, \quad \widetilde{A}^{(r)} = (\mathcal{C}^{(r)})^\top A \mathcal{C}^{(r)},$$

with the constraint that all $\mathcal{C}^{(r)}$ are derived from a single, shared projection $s = \text{HASH}(X)$, thereby ensuring consistency across coarsening levels and enabling adaptive GC.

**Goal 2.** The objective is to learn a coarsening matrix $\mathcal{C} \in \mathbb{R}^{N \times n}$, such that the resulting

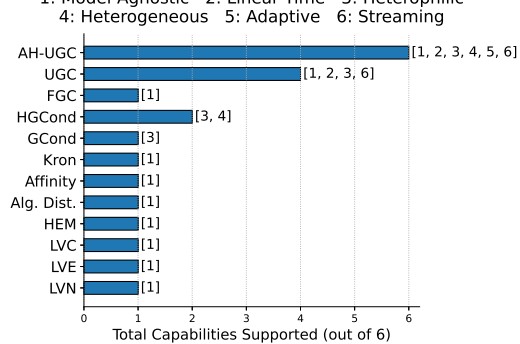

Figure 2: Comparison of capability support across existing GC methods.

coarsened graph $\mathcal{G}_c(\widetilde{V}, \widetilde{E}, \widetilde{\Phi}, \widetilde{\Psi})$ satisfies the following constraints:

$$\widetilde{\Phi}(\widetilde{v}_j) = \Phi(v_i), \quad \forall \widetilde{v}_j \in \widetilde{V}, \forall v_i \in \pi^{-1}(\widetilde{v}_j),$$

$$\widetilde{\Psi}(\widetilde{v}_j, \widetilde{v}_k) = \mathcal{T}_E(i, l) \quad \text{only if} \quad \exists (v_i, v_l) \in E \text{ s.t. } \pi(v_i) = \widetilde{v}_j, \ \pi(v_l) = \widetilde{v}_k,$$

where $\pi : V \to \widetilde{V}$ is the node-to-supernode mapping induced by $\mathcal{C}$. These constraints guarantee that: a) nodes of different types are not merged into the same supernode, and b) edge types between supernodes are consistent with the original heterogeneous schema.

**Related Work.** Graph reduction methods have been extensively studied and can be broadly categorized into optimization-based and GNN-based approaches. Among optimization-driven heuristics, Loukas's spectral coarsening methods (Loukas, 2019) including edge-based (LVE) and neighborhood-based (LVN) variants, preserve the spectral properties of the original graph. Other techniques, such as Heavy Edge Matching (HE)Dhillon et al. (2007); Ron et al. (2010), Algebraic Distance (aJC)Chen & Safro (2011), Affinity GS (aGS) (Livne & Brandt, 2011), and Kron reduction (Dorfler & Bullo, 2013), rely on topological heuristics or structural similarity principles. FGC (Kumar et al., 2023) incorporates node features to learn a feature-aware reduction matrix. Despite their diverse designs, a common drawback of these methods is that they are computationally demanding, often with time complexities ranging from $\mathcal{O}(n^2)$ to $\mathcal{O}(n^3)$, and are not well suited for large-scale or adaptive graph reduction settings. UGC (Kataria et al., 2024), a recent LSH-based framework, addresses these challenges by operating in linear time and supporting heterophilic graphs. However, it produces only a single coarsened graph and must recompute reductions for different coarsening levels, limiting its adaptability. GNN-based condensation methods like GCond (Jin et al., 2021a) and SFGC (Zheng et al., 2024) learn synthetic graphs through gradient matching but require full supervision, are model-specific, and lack scalability. More recent streaming graph condensation methods GECC (Gong et al., 2025) and OpenGC (Gao et al., 2024b) extend this line of work to evolving or open-world settings by updating the condensed graph as new data arrives, but they still operate in a supervised, model-specific regime and do not provide a generic, model-agnostic method. HGCond (Gao et al., 2024a) is designed for heterogeneous graphs, yet it inherits these training and model-dependence limitations and does not support adaptive condensation. Recent works such as CGC (Gao et al., 2025), GCPA (Li et al.), OpenGC (Gao et al., 2024b), and FreeHGC (Liang et al., 2025) propose training-free graph condensation methods. However, CGC, GCPA and OpenGC do not support heterogeneous graphs, and none of these methods are adaptive in nature. Moreover, condensation-based approaches generate synthetic condensed datasets, whereas coarsening operates directly on the original graph, merging nodes while learning representations from the original data. It is worth noting that, while some methods are model-agnostic, others offer partial support for heterophilic or streaming graphs. Yet, no existing approach simultaneously addresses all these challenges: model-agnostic, adaptability, and support for heterophilic, heterogeneous, and streaming graphs. As illustrated in Figure 2, AH-UGC is the first framework to meet all six criteria comprehensively. For details on LSH and consistent hashing, see Appendix C.

**Remark 1** *We provide additional details on the practical applications of graph coarsening in the Appendix D.*

## 3 THE PROPOSED FRAMEWORK: ADAPTIVE AND HETEROGENEOUS UNIVERSAL GRAPH COARSENING

In this section we propose our framework AH-UGC to address the issues of adaptive and heterogeneous graph coarsening. Figure 1 shows the outline of AH-UGC.

### 3.1 ADAPTIVE GRAPH COARSENING(GOAL 1)

The AH-UGC pipeline closely follows the recently proposed structure of UGC Kataria et al. (2024) but incorporates consistent hashing principles to enable adaptive, i.e., multi-level coarsening. Our framework introduces an innovative and flexible approach to graph coarsening that removes the UGC's dependency on fixed bin widths and enables the generation of multiple coarsened graphs. AH-UGC employs an augmented representation to jointly encode both node attributes and graph topology. For a given graph $\mathcal{G}(V, A, X)$, following standard practice in Zhu et al. (2020b); Lim et al. (2021) we compute a heterophily factor $\alpha \in [0, 1]$, which quantifies the relative emphasis on structural information based on label agreement between connected nodes i.e., $\alpha = \frac{|\{(v,u) \in E : y_v = y_u\}|}{|E|}$. This factor is then used to blend node features $X_i$ and adjacency vectors $A_i$. Let $F_i \in \mathbb{R}^d$ denote

the augmented feature vector for node $v_i$. For each node $v_i$ we calculate the augmented feature, $F_i = (1 - \alpha) \cdot X_i \oplus \alpha \cdot A_i$ where $\oplus$ denotes concatenation. This hybrid representation ensures that both local attribute similarity and topological proximity are captured before the coarsening process. Importantly, this design enables our framework to handle heterophilic graphs robustly by incorporating structural properties beyond mere feature similarity.

*Adaptive Coarsening via Consistent and LSH Hashing.* AH-UGC applies $l$ random projection functions using a projection matrix $\mathcal{W} \in \mathbb{R}^{d \times l}$ and bias vector $b \in \mathbb{R}^l$, both sampled from a $p$-stable distribution Indyk & Motwani (1998). The scalar hash score for each projection for $i^{th}$ node is given by:

$$h_i^k = \mathcal{W}_k^\top \cdot F_i + b_k, \quad \forall k \in \{1, \ldots, l\}$$

UGC relies on a bin-width parameter $(r)$ to control the coarsening ratio $(R)$, but determining appropriate bin-widths for different target ratios can be computationally expensive. In contrast, AH-UGC eliminates the need for bin width by leveraging consistent hashing. Once the hash scores $(h_i)$ across projections are computed, AH-UGC enables efficient construction of coarsened graphs at multiple coarsening ratios without requiring reprocessing, making it well-suited for adaptive settings. We define an AGGREGATE function to combine projection scores across multiple random projectors. For each node $i$, the final score $h_i$ is computed as:

$$h_i = \text{AGGREGATE}\left(\left\{h_i^k\right\}_{k=1}^l\right) = \frac{1}{l} \sum_{k=1}^l h_i^k$$

Alternative aggregation functions such as max, median, or weighted averaging can also be used, depending on the design objectives. After computing the scalar hash scores $\{h_i\}$ for all nodes $v_i \in V$, we sort the nodes in increasing order of $h_i$ to form an ordered list $\mathcal{L}$, represented as a list of super-node and mapped nodes: $\mathcal{L} = [\{u_1 : \{v_1\}\}, \{u_2 : \{v_2\}\}, \ldots, \{u_n : \{v_n\}\}]$, where each key $u_j$ denotes a super-node index, and the associated value is the set of nodes currently assigned to that super-node. Initially, each node is its own super-node, and the number of super-nodes is $|V_c^{(0)}| = |V|$. At each iteration $t$, a super-node $u_j$ is randomly selected from the current list $\mathcal{L}^{(t)}$ and merged with its immediate clockwise neighbor $u_{j+1}$. The updated super-node entry is given by:

$$\mathcal{L}^{(t+1)}[j] = \{u_j : \mathcal{L}^{(t)}[u_j] \cup \mathcal{L}^{(t)}[u_{j+1}]\},$$

followed by the removal of $u_{j+1}$ from the list. This reduces the number of super-nodes by one: $|V_c^{(t+1)}| = |V_c^{(t)}| - 1$. The process is repeated until the desired coarsening ratio is reached: $r = \frac{|V_c|}{|V|}$. Furthermore, this coarsening strategy is inherently adaptive, enabling transitions between any two coarsening ratios $r \to t$ directly from the sorted list without reprocessing.

*Construction of Coarsening Matrix $\mathcal{C}$.* Given the score-based node assignments $\pi : V \to \widetilde{V}$, where $\pi[v_i]$ is the super-node index of $v_i$, the binary coarsening matrix $\mathcal{C} \in \{0, 1\}^{N \times n}$ is defined such that $\mathcal{C}_{ij} = 1$ if $\pi[v_i] = \widetilde{v}_j$, and $\mathcal{C}_{ij} = 0$ otherwise. Each entry $\mathcal{C}_{ij}$ of the coarsening matrix is set to 1 if node $v_i$ is assigned to super-node $\widetilde{v}_j$. Since each node receives a unique hash value $h_i$, it is exclusively mapped to a single super-node. This one-to-one assignment guarantees that every super-node has at least one associated node. As a result, each row of $\mathcal{C}$ contains exactly one non-zero entry, ensuring that its columns are mutually orthogonal. The matrix $\mathcal{C}$ therefore adheres to the structural properties defined in Equation 3. The adaptiveness of $\mathcal{C}$ stems from its sensitivity to local projection scores rather than fixed bin constraints.

*Construction of the Coarsened Graph $\mathcal{G}_c$.* The final coarsened graph $\mathcal{G}_c = (\widetilde{V}, \widetilde{A}, \widetilde{F})$ is constructed from the coarsening matrix $\mathcal{C}$. Two super-nodes $\widetilde{v}_i$ and $\widetilde{v}_j$ are connected if there exists at least one edge $(u, v) \in E$ with $u \in \pi^{-1}(\widetilde{v}_i)$ and $v \in \pi^{-1}(\widetilde{v}_j)$. The weighted adjacency matrix is obtained via matrix multiplication: $\widetilde{A} = \mathcal{C}^\top A \mathcal{C}$. The super-node features are computed as the average of the features of the original nodes merged into the super-node: $\widetilde{F}_i = \frac{1}{|\pi^{-1}(\widetilde{v}_i)|} \sum_{u \in \pi^{-1}(\widetilde{v}_i)} F_u$. This ensures that the coarsened representation preserves the aggregate semantic and structural content of its constituent nodes. Since each super-edge aggregates multiple edges from the original graph, $\widetilde{A}$ is significantly sparser than $A$, leading to lower memory and computation requirements downstream. Algorithm 1 in Appendix H outlines the sequence of steps in our AH-UGC framework. The runtime analysis is included in Appendix I.

## 3.2 Heterogeneous Graph Coarsening(Goal 2)

In this section, we present AH-UGC's capability to handle heterogeneous graphs. Given a heterogeneous graph,

$$\mathcal{G}\left(A\{A_{(\text{author, write, paper})},\ A_{(\text{reader, read, paper})}\},\ X\{\mathbf{X}_{(\text{author})},\ X_{(\text{reader})},\ X_{(\text{paper})}\},\ Y\{y_{(\text{paper})}\}\right).$$

AH-UGC proceeds by first partitioning $\mathcal{G}$ by node type and independently applying the coarsening framework to each subgraph. This ensures that only semantically similar nodes are grouped into supernodes and that type-specific structure and features are preserved. Our approach naturally supports varying feature dimensions and allows different coarsening ratios $\eta_{\text{type}}$ across node types. Figure 7 in Appendix J illustrates this process, highlighting how AH-UGC preserves semantic meaning compared to other GC methods that merge heterogeneous nodes indiscriminately.

*Construction of the Coarsened Heterogeneous Graph $\mathcal{G}_c$.* The output of AH-UGC consists of a set of coarsening matrices

$$\mathcal{C}_{\mathcal{H}} = \{\mathcal{C}_{(t)} \in \{0,1\}^{|V_{(t)}| \times |\widetilde{V}_{(t)}|}\}_{t \in \mathcal{T}},$$

each of which maps original nodes of type $t$ i.e., $V_{(t)}$ to their corresponding super-nodes $\widetilde{V}_{(t)}$. Using these mappings, we construct the coarsened graph

$$\mathcal{G}_c\left(\widetilde{A}\{\widetilde{A}_{(\text{author, write, paper})},\ \widetilde{A}_{(\text{reader, read, paper})}\},\ \widetilde{X}\{\widetilde{X}_{(\text{author})},\ \widetilde{X}_{(\text{reader})},\ \widetilde{X}_{(\text{paper})}\},\ \widetilde{Y}\{\widetilde{y}_{(\text{paper})}\}\right).$$

For each node type $t$, the coarsened feature matrix is computed as: $\widetilde{X}_{(t)} = \mathcal{C}_{(t)} \cdot X_{(t)}$, where rows of $\mathcal{C}_{(t)}$ are row-normalized so that super-node features represent the average of their constituent nodes. The label matrix $\widetilde{y}_{(\text{paper})}$ is computed by majority voting over the labels of nodes merged into each super-node. To compute the coarsened edge matrices, for each edge type $\mathcal{T}_e \in \mathcal{T}_E$, we consider the interaction between supernodes of types node-type$_1$ and node-type$_2$, corresponding to the edge relation $e = (\text{node-type}_1, \mathcal{T}_e, \text{node-type}_2) \in \widetilde{E}$. The coarsened adjacency matrix $\widetilde{A}_{(e)}$ is then computed as:

$$\widetilde{A}_{(e)} = \mathcal{C}_{(\text{node-type}_1)} \cdot A_{(e)} \cdot \mathcal{C}_{(\text{node-type}_2)}^{\top}.$$

This formulation accumulates the edge weights between the original nodes to define the inter-supernode connections, thereby preserving the structural connectivity patterns between different node-types of the original graph. Since each edge type is coarsened independently based on the mappings from its corresponding node types, $\mathcal{G}_c$ preserves the heterogeneous semantics and topological relationships of the original graph $\mathcal{G}$. Algorithm 2 in Appendix H outlines the sequence of steps in our AH-UGC framework.

## 3.3 Justification for LSH–CH Supernode Construction.

Since the list $\mathcal{L}$ is constructed using locality-sensitive hashing (LSH) principles Indyk & Motwani (1998), i.e., similar nodes are positioned adjacently. Theorem 3.1 shows that nodes that are close in feature space are mapped to nearby positions under our LSH-based projections, while Lemma 1 bounds the probability that a distant point appears between two such neighbors in the sorted projection order. Hence, when we merge a node with its immediate clockwise neighbor following CH principles, we are merging similar nodes (with high probability) to form a supernode. This locality-preserving merge yields semantically coherent supernodes and provides the algorithmic justification for the CH step. For completeness, we also include an ablation that replaces the immediate neighbor with the $k^{\text{th}}$ rightward neighbor; see Section 4.3. Results show that when a node is asked to look to its right and merge, it is likely to find a similar neighbor and not a random or noisy one.

**Theorem 3.1** *Let* $x, y \in \mathbb{R}^d$, *and let the projection function be defined as:* $h(x) = \sum_{j=1}^{\ell} w_j^{\top} x$, $r_j \sim \mathcal{N}(0, I_d)$ *i.i.d. Then the difference* $h(x) - h(y) \sim \mathcal{N}(0, \ell \|x - y\|^2)$, *and for any* $\varepsilon > 0$:

$$\Pr\left[|h(x) - h(y)| \leq \varepsilon\right] = \text{erf}\left(\frac{\varepsilon}{\sqrt{2\ell}\|x - y\|}\right)$$

*Proof:* The proof is deferred in Appendix E.

This gives the probability that two nodes, initially close in the feature space, are projected within an $\epsilon$-range in the projection space.

**Lemma 1** *Let $x, y, z \in \mathbb{R}^d$, with $\|x - y\| \ll \|x - z\|$. Then the probability that a distant point $z$ lies between $x$ and $y$ after projection is:*

$$\Pr[h(x) < h(z) < h(y)] \leq \Phi \left( \frac{\|x - y\|}{\sqrt{\ell} \|x - z\|} \right)$$

*where $\Phi$ is the cumulative distribution function (CDF) of the standard normal distribution. This result ensures that distant nodes rarely interrupt merge candidates that are close in feature space, preserving the structural consistency of coarsened regions.*

*Proof:* The proof is deferred in Appendix F.

By leveraging consistent hashing, our method ensures balanced supernode formation. Theorem 3.2 provides a probabilistic upper bound on the number of nodes mapped to any supernode.

**Theorem 3.2 (Explicit Load Balance via Random Rightward Merges)** *Let $n$ nodes be sorted according to the consistent hashing scores defined earlier. Let $k$ supernodes be formed by performing $n - k$ random rightward merges in the sorted list. Then, for any constant $c > 0$, the maximum number of nodes in any supernode $S_i$ satisfies:*

$$\Pr \left[ \max_i |S_i| \leq \frac{n}{k} + \frac{n(\log k + c)}{k} \right] \geq 1 - e^{-c}$$

*Proof:* The proof is deferred in Appendix G.

## 4 EXPERIMENTS

We conduct comprehensive experiments to evaluate the effectiveness of AH-UGC. First, we validate its ability to perform *adaptive graph coarsening*. Second, we assess the quality of coarsened graphs using node classification accuracy and spectral similarity. Finally, we demonstrate AH-UGC's generalizability by evaluating its performance on *heterogeneous graphs*. These datasets enable us to evaluate all six key components outlined in Section 2.1. For detailed dataset statistics and System Specifications, refer to Table 7 in Appendix B.

Table 1: Total time (in seconds) to generate coarsened graphs at multiple resolutions, targeting a set of coarsening ratios of $\mathcal{R} = \{0.55, 0.50, 0.45, 0.40, 0.35, 0.30, 0.25, 0.20, 0.15, 0.10\}$. The best and the second-best accuracies in each row are highlighted by dark and lighter shades of Green, respectively. "OOT" indicates out-of-time or memory errors.

| Dataset | VAN | VAE | VAC | HE | aJC | aGS | Kron | FGC | LAGC | UGC | AH-UGC |
|---------|-----|-----|-----|-----|-----|-----|------|-----|------|-----|--------|
| PubMed | 166 | 224 | 510 | 213 | 231 | 2351 | 155 | OOT | OOT | 137 | 29 |
| CS | 174 | 237 | 343 | 216 | 256 | 1811 | 204 | OOT | OOT | 233 | 23 |
| Physics | 411 | 798 | 943 | 705 | 906 | 9341 | 755 | OOT | OOT | 331 | 54 |
| Chameleon | 31 | 17 | 104 | 20 | 32 | 82 | 15 | OOT | OOT | 21 | 6.73 |
| Squirrel | 384 | 61 | 398 | 66 | 342 | 1113 | 68 | OOT | OOT | 53 | 4.69 |
| Film | 64 | 34 | 255 | 36 | 44 | 257 | 30 | OOT | OOT | 92 | 11 |
| Flickr | 1199 | 2301 | 24176 | 2866 | 3421 | 59585 | 2858 | OOT | OOT | 187 | 51 |
| ogbn-arxiv | OOT | OOT | OOT | OOT | OOT | OOT | OOT | OOT | OOT | 1394 | 185 |
| Reddit | OOT | OOT | OOT | OOT | OOT | OOT | OOT | OOT | OOT | 1595 | 290 |
| Yelp | OOT | OOT | OOT | OOT | OOT | OOT | OOT | OOT | OOT | 6904 | 1374 |

### 4.1 ADAPTIVE COARSENING RUN-TIME.

Given a graph $\mathcal{G}$, we evaluate AH-UGC's ability to adaptively coarsen it to multiple resolutions, targeting a set of coarsening ratios $\mathcal{R} = \{0.55, 0.50, 0.45, ...0.10\}$. As described in Section 3, AH-UGC leverages LSH and consistent hashing to group similar nodes into supernodes, enabling the construction of multiple coarsened graphs in a single pass. This adaptivity significantly reduces computational overhead compared to existing methods, which typically require reprocessing the entire graph for each target resolution. The computational advantages of our approach are evident in Table 1 and Table 9 in App. K, where AH-UGC outperforms all baseline methods by a significant margin, achieving the lowest coarsening time across all datasets and coarsening ratios, while maintaining scalability even on large-scale graphs where other methods fail.

Table 2: Illustration of spectral properties preservation, including HE, RcE and REE at 0.5 coarsening ratio.

|  | Dataset | VAN | VAE | VAC | HE | aJC | aGS | Kron | UGC | AH-UGC |
|---|---|---|---|---|---|---|---|---|---|---|
| HE Error | DBLP | 2.20 | 2.07 | 2.21 | 2.21 | 2.12 | 2.06 | 2.24 | 2.10 | 1.99 |
|  | Pubmed | 2.49 | 3.33 | 3.46 | 3.19 | 2.77 | 2.48 | 2.74 | 1.72 | 1.53 |
|  | Squirrel | 4.17 | 2.61 | 2.72 | 1.52 | 1.92 | 2.01 | 1.87 | 0.69 | 0.82 |
| ReC Error | DBLP | 4.94 | 4.89 | 5.03 | 5.06 | 5.03 | 4.73 | 5.08 | 5.24 | 5.11 |
|  | Pubmed | 4.48 | 5.13 | 5.14 | 5.08 | 5.03 | 4.78 | 4.99 | 4.60 | 4.43 |
|  | Squirrel | 10.36 | 9.90 | 10.31 | 9.13 | 9.88 | 10.00 | 9.39 | 9.09 | 9.07 |
| REE Error | DBLP | 0.10 | 0.05 | 0.13 | 0.07 | 0.06 | 0.03 | 0.18 | 0.44 | 0.32 |
|  | Pubmed | 0.05 | 0.97 | 0.88 | 0.71 | 0.48 | 0.06 | 0.42 | 0.31 | 0.21 |
|  | Squirrel | 0.88 | 0.58 | 0.42 | 0.44 | 0.34 | 0.36 | 0.48 | 0.05 | 0.07 |

## 4.2 SPECTRAL PROPERTIES PRESERVATION.

Following the experimental setup of Kumar et al. (2023); Kataria et al. (2024); Loukas (2019) we use Hyperbolic Error (HE), Reconstruction Error (RcE) and Relative Eigen Error (REE) to indicate the structural similarity between $\mathcal{G}$ and $\mathcal{G}_c$. A more detailed discussion about these properties is included in Appendix L. Across three spectral evaluation metrics AH-UGC delivers performance that is comparable to, and in several cases surpasses, state-of-the-art methods, see Table 2. While there are minor dips in performance on a few datasets, this trade-off can be justified given the significant computational efficiency and scalability gains offered by our framework. These results underscore that AH-UGC achieves strong structural fidelity without compromising on runtime, making it especially suitable for large-scale or adaptive coarsening scenarios.

## 4.3 LSH AND CH RESULTS.

We empirically validate Theorem 3.1, see Figure 3 (left). As $\epsilon$ increases, $\Pr[|h(x) - h(y)| \leq \varepsilon]$ approaches 1, consistent with the theoretical erf-based bound. These results justify the use of consistent hashing, where each node is merged with its nearest clockwise neighbor. Theorem 3.1 and Figure 3 (left) guarantee that similar nodes are projected to nearby locations and are thus highly likely to be merged into a supernode. We ablate the "merge the $k$-th rightward neighbor". As seen from Figure 3 (right), performance

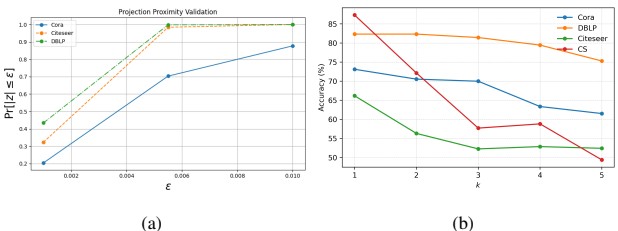

(a)                    (b)

Figure 3: Validation of the LSH–CH design: (a) empirical evidence that nearby feature vectors remain close after LSH projection; (b) node-classification accuracy when merging with the $k$-th rightward neighbor—performance is best at $k{=}1$ and degrades as $k$ increases.

degrades as $k$ increases, aligning with the proposition, i.e., when k=1, the merged node pairs are most likely to be semantically similar, resulting in the highest accuracy and best quality coarsened graph. As k increases, we merge less similar nodes, degrading the representational quality of the graph; this validates Lemma 1.

## 4.4 NODE CLASSIFICATION AND LINK PREDICTION ACCURACY

Graph Neural Networks (GNNs) are widely used for node classification tasks, where the goal is to predict labels for nodes based on both node features and the underlying graph structure. In this context, we evaluate the effectiveness of AH-UGC by examining how well it preserves predictive performance when downstream models are trained on coarsened graphs Huang et al. (2021). Specifically, we train several GNN models on the coarsened version of the original graph while evaluating their performance on the original graph's test nodes. Following established practice in the literature, we employ different GNN backbones tailored to each graph type. For "homophilic" datasets, we use *GCN* Kipf & Welling (2016), *Sage* Hamilton et al. (2017), *GAT* Velickovic et al. (2018), *GIN* Xu et al. (2018a) and *APPNP* Huang et al. (2021), which are well-suited to leverage dense neighborhood similarity. For "heterophilic" datasets, we adopt *GPRGNN* Chien et al. (2020), *MixHop* Abu-El-Haija et al. (2019), *H2GNN* Zhu et al. (2020b), *GCN-II* Chen et al. (2020), *GatJK* Xu et al. (2018b) and *SGC* Wu et al. (2019), which are designed to

Table 3: Node classification accuracy across various datasets and models at 0.5 coarsening ratio.

| Dataset | Model | VAN | VAE | VAC | HE | aJC | aGS | Kron | UGC | AH-UGC | Base |
|---------|-------|-----|-----|-----|----|-----|-----|------|-----|--------|------|
| PubMed | GCN | 85.73 | 86.74 | 86.66 | 87.60 | 86.11 | 86.08 | 86.11 | 84.66 | 85.47 | 87.60 |
| | GIN | 81.98 | 82.07 | 82.78 | 60.11 | 79.03 | 82.96 | 81.49 | 82.42 | 83.97 | 85.75 |
| | GAT | 84.32 | 69.78 | 81.11 | 50.60 | 75.99 | 84.23 | 83.90 | 84.66 | 84.63 | 87.39 |
| Physics | GCN | 94.75 | 94.62 | 94.57 | 94.73 | 94.39 | 94.75 | 94.40 | 95.20 | 94.88 | 95.79 |
| | GIN | 94.90 | 94.56 | 94.78 | 94.49 | 93.79 | 94.79 | 92.65 | 94.41 | 94.94 | 95.66 |
| | GAT | 94.97 | 95.01 | 95.00 | 94.65 | 95.36 | 94.60 | 94.85 | 96.02 | 95.10 | 94.28 |
| Chameleon | SGC | 38.60 | 51.58 | 45.79 | 54.91 | 52.63 | 53.15 | 54.39 | 58.60 | 59.65 | 57.46 |
| | Mixhop | 40.53 | 51.40 | 43.33 | 50.35 | 49.82 | 49.30 | 54.39 | 58.25 | 58.60 | 63.16 |
| | GPR-GNN | 40.53 | 46.32 | 41.05 | 39.64 | 40.35 | 43.68 | 51.05 | 54.74 | 52.28 | 55.04 |
| Penn94 | SGC | 62.93 | 62.33 | 62.23 | 62.13 | 63.52 | 63.03 | 63.52 | 75.74 | 75.87 | 66.78 |
| | Mixhop | 71.71 | 69.62 | 69.35 | 68.36 | 67.98 | 68.40 | 67.98 | 73.36 | 72.13 | 80.28 |
| | GPR-GNN | 68.18 | 68.19 | 68.36 | 68.20 | 67.77 | 68.15 | 68.11 | 67.93 | 68.55 | 79.43 |

handle weak or inverse homophily. For "heterogeneous" graphs, we use *HeteroSGC, HeteroGCN, HeteroGCN2* Gao et al. (2024a) models that respect node and edge types during message passing. Complete architectural and hyperparameter details are provided in Appendix M. Table 3 reports node classification accuracy for homophilic and heterophilic graphs on a representative subset of datasets and GNN models. Please refer to Table 12 in Appendix M for comprehensive results across additional datasets and architectures. The AH-UGC framework consistently delivers results that are either on par with or exceed the performance of existing coarsening methods. As shown in Table 3, the framework is independent of any particular

Table 4: Link prediction accuracy (%).

| Dataset | AH-UGC | UCG | VAN | HE | Kron |
|---------|--------|-----|-----|----|------|
| DBLP | 88.63 | 87.48 | 89.14 | 88.36 | 88.12 |
| Pubmed | 91.84 | 92.78 | 91.81 | 91.45 | 92.05 |
| Squirrel | 91.15 | 91.09 | 91.03 | 93.45 | 92.41 |
| Chameleon | 90.17 | 90.96 | 89.45 | 92.41 | 92.84 |

GNN architecture, highlighting its robustness and model-agnostic characteristics. AH-UGC is not limited to a single downstream task. To further validate the quality of the coarsened graph, we employ the coarsened graph to train a GNN model for the link prediction task. Link prediction accuracy (%) across four datasets; results are summarized in Table 4.

*Performance on Heterogeneous Graphs:* As outlined in Section 3, conventional graph coarsening techniques struggle with preserving the semantic integrity of heterogeneous graphs. In contrast, AH-UGC explicitly enforces type-aware coarsening, ensuring that supernodes are composed of nodes from a single type, thus maintaining the heterogeneity semantics. Table 5 presents node classification accuracies across various heterogeneous GNN models.

AH-UGC consistently outperforms other methods due to its ability to preserve type purity within supernodes. This structural consistency enables all tested GNN architectures to achieve significantly higher classification performance. Figure 5 illustrates the degree of supernode impurity for each method. Each bar corresponds to a supernode and depicts the percentage distribution of node types within it. While supernodes generated by AH-UGC are entirely type-pure, those produced by baseline methods exhibit substantial cross-type mixing, leading to semantic drift and reduced model performance. Figure 4 analyzes the effect of increasing

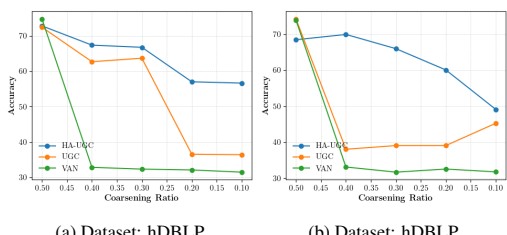

(a) Dataset: hDBLP     (b) Dataset: hDBLP

Figure 4: Node classification accuracy under decreasing coarsening ratios for two heteroGNN models: HeteroGCN and HeteroGCN2.

coarsening ratios on node classification accuracy. As expected, all methods experience performance degradation with aggressive coarsening. However, the drop is exponential for existing approaches due to rising impurity levels. In contrast, AH-UGC maintains structural purity across coarsening levels, resulting in a gradual, near-linear decline in accuracy. This robustness demonstrates AH-UGC's superior capacity to coarsen heterogeneous graphs while preserving their semantic and structural fidelity.

Table 5: Node classification accuracy (%) for heterogeneous datasets at 0.30 coarsening ratio.

| Dataset | Model | VAN | VAE | VAC | HE | aJC | aGS | Kron | UGC | AH-UGC | Base |
|---------|-------|-----|-----|-----|-----|-----|-----|------|-----|--------|------|
| IMDB | HeteroSGC | 30.53 | 27.82 | 27.42 | 27.42 | 27.42 | 27.30 | 27.42 | 49.61 | 51.46 | 66.74 |
| | HeteroGCN | 35.40 | 36.36 | 35.82 | 35.46 | 35.7 | 35.7 | 35.93 | 47.84 | 52.91 | 61.72 |
| | HeteroGCN2 | 36.13 | 36.15 | 35.82 | 35.82 | 35.82 | 35.82 | 35.82 | 44.13 | 52.58 | 63.47 |
| DBLP | HeteroSGC | 28.33 | 28.33 | 29.43 | 53.07 | 54.65 | 29.43 | 29.43 | 53.92 | 56.60 | 94.10 |
| | HeteroGCN | 32.07 | 31.08 | 32.75 | 32.75 | 33 | 35.46 | 31.28 | 58.82 | 63.13 | 84.18 |
| | HeteroGCN2 | 31.33 | 31.35 | 31.77 | 33.25 | 31.12 | 32.01 | 32.63 | 58.18 | 62.71 | 79.33 |
| ACM | HeteroSGC | 74.25 | 44.66 | OOT | 34.54 | 42.31 | 34.54 | 42.31 | 60.33 | 53.82 | 92.06 |
| | HeteroGCN | 36.33 | 37.65 | OOT | 35.7 | 35.2 | 35.53 | 35.1 | 39.27 | 85.16 | 92.72 |
| | HeteroGCN2 | 36.76 | 34.64 | OOT | 36.19 | 37.35 | 35.04 | 37.35 | 49.62 | 84.36 | 92.72 |



Figure 5: Supernode impurity across AH-UGC (left), UGC (center) and VAN (right) on IMDB dataset. Different colors represent different node types(*Movie, Director, Actor*).

## 5 CONCLUSION

In this paper, we propose AH-UGC, a unified framework for adaptive and heterogeneous graph coarsening. By integrating Locality-Sensitive Hashing (LSH) with Consistent Hashing, AH-UGC efficiently produces multiple coarsened graphs with minimal overhead. Additionally, its type-aware design ensures semantic preservation in heterogeneous graphs by avoiding cross-type node merges. The framework is model-agnostic, scalable, and capable of handling both heterophilic and heterogeneous graphs. We demonstrate that AH-UGC preserves key spectral properties, making it applicable across diverse graph types. Extensive experiments on 23 real-world datasets with various GNN architectures show that AH-UGC consistently outperforms existing methods in scalability, classification accuracy, and structural fidelity.

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

# Appendix

## A   NOTATIONS.

Below we summarize the main notation and abbreviations used in this paper.

Table 6: Notation and abbreviations used in the paper.

| Symbols and abbreviations | Description |
| --- | --- |
| GC | Graph coarsening |
| AH-UGC | Adaptive heterogeneous universal graph coarsening |
| GNN | Graph neural network |
| HGNN | Heterogeneous graph neural network |
| LSH | Locality-Sensitive Hashing |
| CH | Consistent Hashing |
| UGC | Universal Graph Coarsening |
| VAN | Local Variation Neighbourhood |
| VAE | Local Variation Edge |
| VAC | Local Variation Clique |
| HE | Heavy Edge Matching |
| aJC | Algebraic Distance |
| aGS | Affinity GS |
| FGC | Featured Graph Coarsening |
| $\mathcal{G}$ | Graph |
| $V$ | Set of vertices |
| $E$ | Set of edges |
| $A$ | Adjacency matrix of $\mathcal{G}$ |
| $X$ | Node feature matrix |
| $D$ | Degree matrix |
| $L$ | Graph Laplacian matrix |
| $\mathcal{G}(V, A, X)$ | Graph given by vertices, adjacency, and features |
| $\mathcal{G}(L, X)$ | Graph represented by Laplacian and features |
| $N$ | Number of nodes |
| $m$ | Number of edges |
| $\Phi$ | Node-type map |
| $\Psi$ | Edge-type map |
| $T_V$ | Set of node types |
| $T_E$ | Set of edge / relationship types |
| $\mathcal{G}(V, E, \Phi, \Psi)$ | Entity-based heterogeneous graph |
| $\mathcal{G}(X_{node\_type}, A_{edge\_type}, y_{target\_type})$ | Type-based heterogeneous graph |
| $r$ | Coarsening ratio |
| $\mathcal{C}$ | Coarsening matrix |
| $\mathcal{C}^{(r)}$ | Coarsening matrix at ratio $r$ |
| $\mathcal{G}_c$ | Coarsened graph |
| $\tilde{V}$ | Set of supernodes |
| $A_c$ | Coarsened adjacency matrix |
| $X_c$ | Coarsened node feature matrix |
| $\alpha$ | Heterophily factor |
| $\mathcal{W}$ | Projection matrix |
| $F$ | Augmented feature matrix |
| $\pi$ | Node-to-supernode mapping |
| $h_i$ | Hash score for node $i$ derived from features |
| $\mathcal{R}$ | List of coarsening ratios |
| $\mathcal{L}^t$ | Ordered list of hash values at timestamp $t$ |
| $\mathcal{C}_{\mathcal{H}}$ | Set of coarsening matrices for heterogeneous graphs |

## B DATASETS

We experiment on 24 widely-used benchmark datasets grouped into four categories: **(a) Homophilic**: *Cora ,Citeseer, Pubmed* Yang et al. (2016), *CS, Physics* (Shchur et al., 2018), *DBLP* (Fu et al., 2020); **(b) Heterophilic**: *Squirrel, Chameleon, Texas, Cornell, Film, Wisconsin* Zhu et al. (2020a); Pei et al. (2020); Zhu et al. (2021); Du et al. (2022), *Penn49, deezer-europe, Amherst41, John Hopkins55, Reed98* Lim et al. (2021); **(c) Heterogeneous**: *IMDB, DBLP, ACM* Liu et al. (2023a); Gao et al. (2024a); and **(d) Large-scale**: *Flickr, Yelp*, Zeng et al. (2019) *ogbn-arxiv* Wang et al. (2020) , *Reddit* Hamilton et al. (2017). These datasets enable us to evaluate all six key components outlined in Section 2.1. Please refer to Table 7 and 8 for detailed dataset statistics and characteristics.

**System Specifications:** All experiments are conducted on a server equipped with two **NVIDIA RTX A6000** GPUs (48 GB memory each) and an **Intel Xeon Platinum 8360Y** CPU with **1 TB RAM**.

Table 7: Summary of the datasets.

| Category | Data | Nodes | Edges | Feat. | Class | H.R($\alpha$) |
|---|---|---|---|---|---|---|
| Homophilic dataset | Cora | 2,708 | 5,429 | 1,433 | 7 | 0.19 |
| | Citeseer | 3,327 | 9,104 | 3,703 | 6 | 0.26 |
| | DBLP | 17,716 | 52,867 | 1,639 | 4 | 0.18 |
| | CS | 18,333 | 163,788 | 6,805 | 15 | 0.20 |
| | PubMed | 19,717 | 44,338 | 500 | 3 | 0.20 |
| | Physics | 34,493 | 247,962 | 8,415 | 5 | 0.07 |
| Heterophilic dataset | Texas | 183 | 309 | 1703 | 5 | 0.91 |
| | Cornell | 183 | 295 | 1703 | 5 | 0.70 |
| | Film | 7600 | 33544 | 931 | 5 | 0.78 |
| | Squirrel | 5201 | 217073 | 2089 | 5 | 0.78 |
| | Chameleon | 2277 | 36101 | 2325 | 5 | 0.75 |
| | Penn94 | 41,554 | 1.36M | 5 | 2 | 0.53 |
| | Deezer-europe | 28,281 | 185.5k | 31.24k | 2 | - |
| | Amherst41 | 2235 | 181.9k | 1193 | 3 | - |
| | John-Hopkin55 | 41,554 | 2.7M | 4,814 | 3 | - |
| | Reed98 | 962 | 37.6k | 745 | 3 | - |
| Large dataset | Flickr | 89,250 | 899,756 | 500 | 7 | - |
| | Reddit | 232,965 | 11.60M | 602 | 41 | - |
| | Ogbn-arxiv | 169,343 | 1.16M | 128 | 40 | - |
| | Yelp | 716,847 | 13.95M | 300 | 100 | - |

Table 8: Summary of Heterogeneous graph datasets

| Dataset | Nodes | Edges | Features | Classes |
|---|---|---|---|---|
| IMDB | Movie - 4278
Director - 2081
Actor - 5257 | (Movie, to, Director) - 4278
(Movie, to, Actor) - 12828
(Director, to, Movie) - 4278
(Actor, to, Movie) - 12828 | 3061 | Movie: 3 |
| DBLP | Author - 4057
Paper - 4231
Term - 7723
Conference - 50 | (Author, to, Paper) - 19645
(Paper, to, Author) - 19645
(Paper, to, Term) - 85810
(Paper, to, Conference) - 14328
(Term, to, Paper) - 85810
(Conference, to, Paper) - 14328 | Author - 334
Paper - 4231
Term - 50
Conference - NA | Author: 4 |
| ACM | Paper - 3025
Author - 5959
Subject - 56
Term - 1902 | (Paper, cite, Paper) - 5343
(Paper, ref, Paper) - 5343
(Paper, to, Author) - 9949
(Author, to, Paper) - 9949
(Paper, to, Subject) - 3025
(Subject, to, Paper) - 3025
(Paper, to, Term) - 255619
(Term, to, Paper) - 255619 | All except term - 1902
Term - NA | Paper: 3 |

## C LOCALITY-SENSITIVE HASHING AND CONSISTENT HASHING

Locality-Sensitive Hashing (LSH) is a technique for hashing high-dimensional data points so that similar items are more likely to collide (i.e., hash to the same bucket) Indyk & Motwani (1998); Kulis & Grauman (2009); Buhler (2001). It is commonly used in approximate nearest neighbor search, dimensionality reduction, and randomized algorithms Chum et al. (2007). For example, a hash function $h(\cdot)$ is locality-sensitive with respect to a similarity measure $s(\cdot, \cdot)$ if $\Pr[h(x) =$

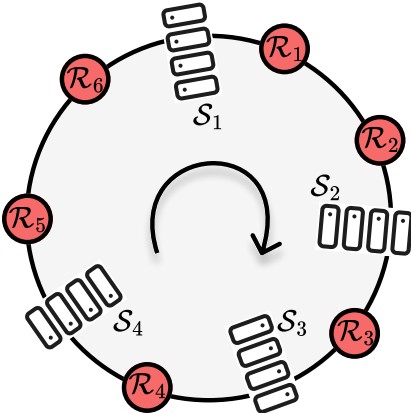

Figure 6: Consistent Hashing (CH): Objects and bins are hashed to a unit circle; each object is assigned to the next bin in clockwise order.

$h(y)]$ increases with $s(x, y)$. Gaussian LSH schemes, such as those using random projections, are particularly effective for preserving Euclidean distances Kataria et al. (2023; 2024).

In the consistent hashing (CH) Karger et al. (1997); Chen et al. (2021) scheme, objects/requests are hashed to random bins/servers on the unit circle, as shown in Figure 6. Objects are then assigned to the closest bin in the clockwise direction. CH was originally proposed for load balancing in distributed systems; it maps data points to buckets such that small changes in input (e.g., adding or removing an object) do not drastically affect the overall assignment. We aim to employ CH for adaptive graph coarsening, as it enables stable and scalable grouping of similar objects/nodes. When combined with LSH, consistent hashing offers a powerful mechanism for adaptive graph reduction.

## D PRACTICAL IMPACTS OF GRAPH COARSENING.

We would like to emphasize that graph coarsening is not solely motivated by GPU memory reduction during GNN training. Instead, coarsening serves as a fundamental preprocessing technique that enables scalable, interpretable, and efficient graph learning in large-scale and dynamic settings. While memory reduction is one practical application, our use of node classification serves as a proxy task to evaluate the structural quality of the coarsened graphs. Some of the key benefits of graph coarsening and other graph reduction techniques are as follows:

1. **Neural Architecture Search (NAS)**: Graph coarsening/reduction reduces dataset size, enabling faster NAS by minimizing the need to train on full large-scale graphs. This accelerates model selection and lowers compute costs Yang et al. (2023).
2. **Continual Learning**: Informative coarsened graphs act as memory-efficient replay buffers that mitigate catastrophic forgetting in continual learning. For example, CaT Liu et al. (2023b) uses graph reduction for task updates.
3. **Visualization and Explanation**: Smaller graphs are easier to visualize and interpret. Coarsening enables faster and more human-friendly multilevel graph visualization pipelines Zhao et al. (2018).
4. **Privacy Preservation**: Reduced graphs offer inherent privacy benefits by obfuscating fine-grained details. Methods like coarsening/sparsification have been shown to approximate differential privacy while preserving utility Dong et al. (2022).
5. **Graph Data Augmentation**: Coarsening at multiple levels produces diverse graph views, useful for augmentation. For example, HARP generates multi-resolution embeddings via progressive coarsening Zhao et al. (2022).
6. **Low-Memory Deployment**: Compact coarsened graphs can be used to train or infer with GNNs on memory-constrained devices, facilitating edge deployment and mobile graph learning.
7. **Coarsening Applications in Different Domains**:
   - **Biology**: Coarsening has been effectively used to analyze massive single-cell datasets in genomics and cytometry, where full-resolution graphs are computationally prohibitive Kataria et al. (2025).
   - **Chemistry**: By reducing the size of high-fidelity quantum datasets through locality-sensitive hashing, graph coarsening techniques enable the efficient development of accurate ML potentials

for complex chemical systems, significantly lowering the cost of quantum chemical simulations Anmol et al. (2025).

Due to these wide-ranging benefits, **graph coarsening and other reduction techniques remain an active and evolving area of research**. We refer the reviewer to the comprehensive survey Hashemi et al. (2024) for further details.

## E   PROOF OF THEOREM 3.1

**Theorem E.1 (Projection Proximity for Similar Points)** *Let $x, y \in \mathbb{R}^d$, and define the projection function:*

$$h(x) = \sum_{j=1}^{\ell} w_j^\top x, \quad r_j \sim \mathcal{N}(0, I_d) \ i.i.d.$$

*Then the difference $h(x) - h(y) \sim \mathcal{N}(0, \ell \|x - y\|^2)$, and for any $\varepsilon > 0$:*

$$\Pr\left[|h(x) - h(y)| \leq \varepsilon\right] = \mathrm{erf}\left(\frac{\varepsilon}{\sqrt{2\ell}\|x - y\|}\right)$$

**Proof** Let $z = x - y \in \mathbb{R}^d$. Then:

$$h(x) - h(y) = \sum_{j=1}^{\ell} w_j^\top x - \sum_{j=1}^{\ell} w_j^\top y = \sum_{j=1}^{\ell} w_j^\top (x - y) = \sum_{j=1}^{\ell} w_j^\top z$$

Each term $r_j^\top z$ is a linear projection of a standard Gaussian vector, hence:

$$r_j^\top z \sim \mathcal{N}(0, \|z\|^2) = \mathcal{N}(0, \|x - y\|^2)$$

Since the $r_j$ are independent, the sum of $\ell$ such independent variables is:

$$h(x) - h(y) \sim \mathcal{N}(0, \ell \|x - y\|^2)$$

Now consider the probability:
$$\Pr\left[|h(x) - h(y)| \leq \varepsilon\right]$$

This is the cumulative probability within $\varepsilon$ of a zero-mean Gaussian with variance $\ell \|x - y\|^2$. Let $\sigma^2 = \ell \|x - y\|^2$. Then:

$$\Pr\left[|Z| \leq \varepsilon\right] = \mathrm{erf}\left(\frac{\varepsilon}{\sqrt{2\sigma^2}}\right) = \mathrm{erf}\left(\frac{\varepsilon}{\sqrt{2\ell}\|x - y\|}\right)$$

as required.

## F   PROOF OF LEMMA 1

**Lemma** Let $x, y, z \in \mathbb{R}^d$, with $\|x - y\| \ll \|x - z\|$. Then the probability that a distant point $z$ lies between $x$ and $y$ after projection is:

$$\Pr[h(x) < h(z) < h(y)] \leq \Phi\left(\frac{\|x - y\|}{\sqrt{\ell}\|x - z\|}\right)$$

where $\Phi$ is the cumulative distribution function (CDF) of the standard normal distribution. This result ensures that distant nodes rarely interrupt merge candidates that are close in feature space, preserving the structural consistency of coarsened regions.

**Proof** We analyze the chance that a far-away point $z$ lies between two close points $x$ and $y$ in the projected order.

Let:
$$a = h(x) = \sum_j w_j^\top x, \qquad b = h(y), \qquad c = h(z).$$

Define the difference $d = h(y) - h(x) \sim \mathcal{N}\left(0, \ell \|x - y\|^2\right)$.

Assume without loss of generality that $h(x) < h(y)$. Then:

$$\Pr\left[h(x) < h(z) < h(y)\right] \;=\; \Pr\left[c - a \in (0, d]\right].$$

Since $h(z) - h(x) \sim \mathcal{N}\left(0, \ell\|x - z\|^2\right)$, we compute:

$$\Pr\left[0 < h(z) - h(x) < d\right] = \int_0^d \frac{1}{\sqrt{2\pi\,\ell}\,\|x - z\|}\,\exp\left(-\frac{t^2}{2\ell\|x - z\|^2}\right)\,dt \;\leq\; \Phi\left(\frac{d}{\sqrt{\ell}\,\|x - z\|}\right).$$

Taking expectation over $(d)$, this gives the desired bound.

## G    PROOF OF THEOREM 3.2

**Theorem G.1 (Explicit Load Balance via Random Rightward Merges)** *Let $n$ nodes be sorted according to the consistent hashing scores defined earlier. Let $k$ supernodes be formed by performing $n - k$ random rightward merges in the sorted list. Then, for any constant $c > 0$, the maximum number of nodes in any supernode $S_i$ satisfies:*

$$\Pr\left[\max_i |S_i| \leq \frac{n}{k} + \frac{n(\log k + c)}{k}\right] \geq 1 - e^{-c}$$

**Proof** Let $U_1, \ldots, U_{k-1} \sim \text{Uniform}(0, 1)$ and let $U_{(1)} < \cdots < U_{(k-1)}$ be their order statistics. Define the spacings:

$$I_1 = U_{(1)} - 0, \quad I_2 = U_{(2)} - U_{(1)}, \quad \ldots, \quad I_k = 1 - U_{(k-1)}$$

Then $(I_1, \ldots, I_k)$ form a random partition of the unit interval $[0, 1]$. It is a classical result (e.g., David & Nagaraja (2004)) that:

- The vector $(I_1, \ldots, I_k) \sim \text{Dirichlet}(1, \ldots, 1)$,
- Each individual spacing $I_i \sim \text{Beta}(1, k - 1)$.

**Tail bound on $I_i$.** The PDF of $I_i$ is:

$$f(t) = (k - 1)(1 - t)^{k-2}, \quad t \in [0, 1]$$

and its tail probability is:

$$\Pr[I_i > t] = (1 - t)^{k-1}$$

Choose $t = \frac{\log k + c}{k}$. Then:

$$\Pr[I_i > t] \leq \exp\left(-(\log k + c)\right) = \frac{1}{k} e^{-c}$$

**Union bound.** Over all $k$ intervals:

$$\Pr\left[\max_i I_i > \frac{\log k + c}{k}\right] \leq k \cdot \frac{1}{k} e^{-c} = e^{-c} \Rightarrow \Pr\left[\max_i I_i \leq \frac{\log k + c}{k}\right] \geq 1 - e^{-c}$$

**Scaling to $n$ nodes.** We model the sorted list of $n$ nodes as uniformly spaced over $[0, 1]$. Each spacing $I_i$ then corresponds to a fraction of the list, and multiplying by $n$ yields the expected number of nodes in that supernode:

$$|S_i| = n \cdot I_i \Rightarrow \max_i |S_i| = n \cdot \max_i I_i \leq \frac{n}{k} + \frac{n(\log k + c)}{k}$$

This completes the proof.

## H    ALGORITHMS

Algorithm 1 and 2 outlines the sequence of steps for both adaptive and heterogeneous graph coarsening.

---

**Algorithm 1** AH-UGC: Adaptive Universal Graph Coarsening

---

**Require:** Input $\mathcal{G}(V, A, X)$, $l \leftarrow$ Number of Projectors

1: $\alpha = \frac{|\{(v,u) \in E : y_v = y_u\}|}{|E|}$; $\alpha$ is heterophily factor, $y_i \in \mathbb{R}^N$ is node labels, $E$ denotes edge list

2: $F = \{(1 - \alpha) \cdot X \oplus \alpha \cdot A\}$

3: $\mathcal{S} \leftarrow F \cdot \mathcal{W} + b; \mathcal{S} \in \mathbb{R}^{n \times l}$      // compute projections

4: $\mathcal{W} \in \mathbb{R}^{d \times l}, \ b \in \mathbb{R}^l \sim \mathcal{D}(\cdot)$      // sample projections

5: $\mathcal{S} \leftarrow F \cdot \mathcal{W} + b; \ \mathcal{S} \in \mathbb{R}^{n \times l}$      // compute projections

6: $s_i \leftarrow \text{AGGREGATE}(\{\mathcal{S}_{i,k}\}_{k=1}^l) = \frac{1}{l} \sum_{k=1}^l \mathcal{S}_{i,k} \quad \forall i \in \{1, \ldots, n\}$      // mean aggregation

7: $\mathcal{L} \leftarrow \text{sort}(\{v_i\}_{i=1}^n)$ by ascending $s_i$      // ordered node list

8: $\mathcal{L} \leftarrow [\{u_1 : \{v_1\}\}, \{u_2 : \{v_2\}\}, \ldots, \{u_n : \{v_n\}\}]$      // initial super-nodes

9: **while** $|\mathcal{L}|/|V| > r$ **do**

10:      $u_j \sim \text{Uniform}(\mathcal{L})$      // sample a super-node

11:      $\mathcal{L}[u_j] \leftarrow \mathcal{L}[u_j] \cup \mathcal{L}[u_{j+1}]$      // merge with right neighbor

12:      $\mathcal{L} \leftarrow \mathcal{L} \setminus \{u_{j+1}\}$      // remove right neighbor

13: **end while**

14: $\mathcal{C} \in \{0, 1\}^{|\mathcal{L}| \times |V|}, \ \mathcal{C}_{ij} \leftarrow \begin{cases} 1 & \text{if } v_j \in \mathcal{L}[u_i] \\ 0 & \text{otherwise} \end{cases}$      // partition matrix

15: $\mathcal{C} \leftarrow \text{row-normalize}(\mathcal{C})$      // normalize rows: $\sum_j \mathcal{C}_{ij} = 1$

16: $\widetilde{F} \leftarrow \mathcal{C}F \ \ ; \ \ \widetilde{A} \leftarrow \mathcal{C}A\mathcal{C}^\top$      // coarsened features and adjacency

17: **return** $\mathcal{G}_c = (\widetilde{V}, \widetilde{A}, \widetilde{F}), \ \mathcal{C}$

---

**Algorithm 2** Heterogeneous Graph Coarsening

---

**Require:** Graph $\mathcal{G}\left(\{X_{(\text{node\_type})}\}, \{A_{(\text{edge\_type})}\}, \{y_{(\text{target\_type})}\}\right)$, compression ratio $\eta$

**Ensure:** Condensed graph $\mathcal{G}_c\left(\{\widetilde{X}_{(\text{node\_type})}\}, \{\widetilde{A}_{(\text{edge\_type})}\}, \{\widetilde{Y}_{(\text{target\_type})}\}\right)$

1: **for** each node type $t$ **do**

2:      $r_t \leftarrow \eta \cdot |V_t|$

3:      $\mathcal{G}_t^{\text{coarse}}, \mathcal{C}_t \leftarrow \text{AH-UGC}(X_t, A_t, r_t)$

4:      $\widetilde{X}_t \leftarrow$ node features from $\mathcal{G}_t^{\text{coarse}}$

5:      **if** $t$ is target type **then**

6:          $\widetilde{y}_t[i] \leftarrow$ majority vote of $y_j$ for $v_j \in \mathcal{C}_t[i]$

7:      **end if**

8: **end for**

9: **for** each edge type $e = (t_1, t_2)$ **do**

10:      Initialize $\widetilde{A}_e \in \mathbb{R}^{|\widetilde{V}_{t_1}| \times |\widetilde{V}_{t_2}|}$

11:      **for** each $(v_i, v_j) \in A_e$ **do**

12:          $u \leftarrow$ super-node index of $v_i$ via $\mathcal{C}_{t_1}$

13:          $v \leftarrow$ super-node index of $v_j$ via $\mathcal{C}_{t_2}$

14:          $\widetilde{A}_e[u, v] \leftarrow \widetilde{A}_e[u, v] + 1$

15:      **end for**

16: **end for**

17: **return** $\mathcal{G}_c\left(\{\widetilde{X}_{(\text{node\_type})}\}, \{\widetilde{A}_{(\text{edge\_type})}\}, \{\widetilde{Y}_{(\text{target\_type})}\}\right)$

---

## I   RUN TIME ANALAYSIS

For UGC, the original paper shows that one run of the algorithm costs $\mathcal{O}(NLd + m)$, with $N = |V|$, $m = |E|$, $d$ the feature dimension, and $L$ the number of projectors.

However, in the adaptive setting we need to find a suitable bin-width for each target coarsening ratio via the recursive *Bin-width Finder* algorithm of UGC. If this procedure is called $K$ times on average, then cost per ratio becomes $O\big((1 + K)(NLd + m)\big)$. If we require $|R|$ different coarsening ratios, the overall cost is

$$T_{\text{UGC, adaptive}} = \mathcal{O}\big(|R|(1 + K)(NLd + m)\big),$$

since both the bin-width search and the coarsened adjacency construction are repeated for every $r \in R$.

In contrast, AH-UGC performs LSH projections only once in $\mathcal{O}(NLd + m)$, then sorts the hash scores once in $\mathcal{O}(N \log N)$, and finally generates all $|R|$ coarsened graphs by a single pass over

the ordered list in $\mathcal{O}(N(1-r))$ where $N(1-r)$ is the number of node reduced to get the smallest coarsened graph. The total cost is therefore

$$T_{\text{AH-UGC, Adaptive}} = \mathcal{O}\big(NLd + m + N\log N + N(1-r))\big).$$

Thus, for a single coarsening level, AH-UGC and UGC are both near-linear, but for adaptive multi-resolution coarsening, UGC pays the expensive $\mathcal{O}(NLd + m)$ cost $(1+K)|R|$ times, whereas AH-UGC pays it once and then adds only $\mathcal{O}(N\log N + N(1-r))$, making AH-UGC asymptotically cheaper in the adaptive setting.

## J HETEROGENOUS GRAPH COARSENING

Figure 7 illustrates this process, highlighting how AH-UGC preserves semantic meaning compared to other GC methods that merge heterogeneous nodes indiscriminately.

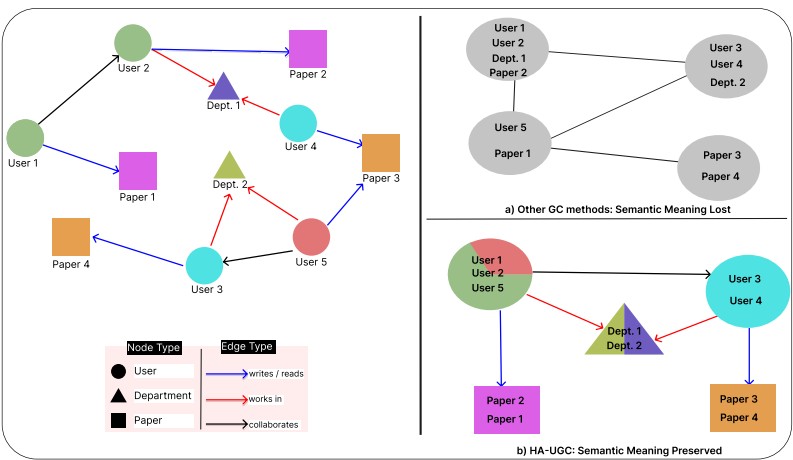

Figure 7: This figure illustrates this process, highlighting how AH-UGC preserves semantic meaning compared to other GC methods that merge heterogeneous nodes indiscriminately.

## K RUN TIME RESULTS

Table 9: Total time (in seconds) to generate coarsened graphs at multiple resolutions, targeting a set of coarsening ratios of $\mathcal{R} = \{0.55, 0.50, 0.45, 0.40, 0.35, 0.30, 0.25, 0.20, 0.15, 0.10\}$. The best and the second-best accuracies in each row are highlighted by dark and lighter shades of Green, respectively. "OOT" indicates out-of-time or memory errors.

| Dataset | VAN | VAE | VAC | HE | aJC | aGS | Kron | FGC | LAGC | UGC | AH-UGC |
|---|---|---|---|---|---|---|---|---|---|---|---|
| Cora | 19 | 13 | 29 | 9 | 13 | 30 | 9 | OOT | OOT | 30 | 7 |
| Citeseer | 28 | 23 | 37 | 21 | 22 | 31 | 20 | OOT | OOT | 28 | 6 |
| DBLP | 162 | 138 | 388 | 204 | 206 | 1270 | 184 | OOT | OOT | 131 | 20 |
| Texas | 1.59 | 0.91 | 2.66 | 0.77 | 0.96 | 1.32 | 0.8 | OOT | OOT | 11 | 0.73 |
| Cornell | 1.76 | 0.99 | 2.72 | 0.86 | 1.11 | 1.35 | 0.68 | OOT | OOT | 9 | 0.79 |

## L SPECTRAL PROPERTIES

1. **Relative Eigen Error (REE):** REE used in Kumar et al. (2023); Kataria et al. (2024); Loukas (2019) gives the means to quantify the measure of the eigen properties of the original graph $\mathcal{G}$ that are preserved in coarsened graph $\mathcal{G}_c$.

   **Definition 4** *REE is defined as follows:*

$$REE(L, L_c, k) = \frac{1}{k}\sum_{i=1}^{k}\frac{|\widetilde{\lambda}_i - \lambda_i|}{\lambda_i} \quad (1)$$

Table 10: This table illustrates spectral properties including HE, RcE, REE across datasets and methods at 50% coarsening ratio. AH-UGC achieves competitive performance across most datasets.

| | Dataset | VAN | VAE | VAC | HE | aJC | aGS | Kron | UGC | AH-UGC |
|---|---|---|---|---|---|---|---|---|---|---|
| HE Error | Cora | 2.04 | 2.08 | 2.14 | 2.19 | 2.13 | 1.95 | 2.14 | 1.96 | 2.03 |
| | DBLP | 2.20 | 2.07 | 2.21 | 2.21 | 2.12 | 2.06 | 2.24 | 2.10 | 1.99 |
| | Pubmed | 2.49 | 3.33 | 3.46 | 3.19 | 2.77 | 2.48 | 2.74 | 1.72 | 1.53 |
| | Squirrel | 4.17 | 2.61 | 2.72 | 1.52 | 1.92 | 2.01 | 1.87 | 0.69 | 0.82 |
| | Chameleon | 2.77 | 2.55 | 2.99 | 1.80 | 1.86 | 1.97 | 1.86 | 1.28 | 1.71 |
| | Deezer-Europe | 1.90 | 1.97 | 2.04 | 1.95 | 1.90 | 1.62 | 1.90 | 1.76 | 1.61 |
| | Penn94 | 1.96 | 1.52 | 1.65 | 1.57 | 1.51 | 1.43 | 1.55 | 1.05 | 1.09 |
| ReC Error | Cora | 3.78 | 3.83 | 3.90 | 3.95 | 3.91 | 3.71 | 3.92 | 4.07 | 4.14 |
| | DBLP | 4.94 | 4.89 | 5.03 | 5.06 | 5.03 | 4.73 | 5.08 | 5.24 | 5.11 |
| | Pubmed | 4.48 | 5.13 | 5.14 | 5.08 | 5.03 | 4.78 | 4.99 | 4.60 | 4.43 |
| | Squirrel | 10.36 | 9.90 | 10.31 | 9.13 | 9.88 | 10.00 | 9.39 | 9.09 | 9.07 |
| | Chameleon | 7.90 | 7.72 | 8.05 | 7.55 | 7.52 | 7.58 | 7.13 | 7.40 | 7.16 |
| | Deezer-Europe | 5.08 | 5.06 | 5.19 | 5.04 | 5.04 | 4.68 | 5.01 | 8.03 | 8.05 |
| | Penn94 | 7.77 | 7.71 | 7.77 | 7.73 | 7.73 | 7.63 | 7.76 | 7.71 | 7.74 |
| REE Error | Cora | 0.09 | 0.07 | 0.05 | 0.04 | 0.11 | 0.09 | 0.03 | 0.64 | 0.66 |
| | DBLP | 0.10 | 0.05 | 0.13 | 0.07 | 0.06 | 0.03 | 0.18 | 0.44 | 0.32 |
| | Pubmed | 0.05 | 0.97 | 0.88 | 0.71 | 0.48 | 0.06 | 0.42 | 0.31 | 0.21 |
| | Squirrel | 0.88 | 0.58 | 0.42 | 0.44 | 0.34 | 0.36 | 0.48 | 0.05 | 0.07 |
| | Chameleon | 0.76 | 0.69 | 0.67 | 0.38 | 0.38 | 0.35 | 0.52 | 0.09 | 0.12 |
| | Deezer-Europe | 0.48 | 0.29 | 0.47 | 0.25 | 0.21 | 0.02 | 0.19 | 0.35 | 0.35 |
| | Penn94 | 0.31 | 0.02 | 0.05 | 0.02 | 0.09 | 0.05 | 0.08 | 0.22 | 0.23 |

where $\lambda_i$ and $\widetilde{\lambda}_i$ are top $k$ eigenvalues of original graph Laplacian $(L)$ and coarsened graph Laplacian $(L_c)$ matrix, respectively.

2. **Hyperbolic error (HE):** HE Bravo Hermsdorff & Gunderson (2019) indicates the structural similarity between $\mathcal{G}$ and $\mathcal{G}_c$ with the help of a lifted matrix along with the feature matrix $X$ of the original graph.
   **Definition 5** *HE is defined as follows:*

$$HE = arccosh(\frac{||(L - L_{\text{lift}})X||_F^2||X||_F^2}{2trace(X^\top LX)trace(X^\top L_{\text{lift}}X)} + 1) \quad (2)$$

   where $L$ is the Laplacian matrix and $X \in \mathbb{R}^{N \times d}$ is the feature matrix of the original input graph, $L_{\text{lift}}$ is the lifted Laplacian matrix defined in Loukas (2019) as $L_{\text{lift}} = \mathcal{C}L_c\mathcal{C}^\top$ where $\mathcal{C} \in \mathbb{R}^{N \times n}$ is the coarsening matrix and $L_c$ is the Laplacian of $\mathcal{G}_c$.

3. **Reconstruction Error (RcE)**
   **Definition 6** *Let $L$ be the original Laplacian matrix and $L_{lift}$ be the lifted Laplacian matrix, then the reconstruction error (RE) Liu et al. (2018); Kumar et al. (2023) is defined as:*

$$RcE = \|L - L_{lift}\|_F^2 \quad (3)$$

## M  NODE CLASSIFICATION ACCURACY

Graph Neural Networks (GNNs), designed to operate on graph data Kataria et al. (2024); Malik et al.

Table 11: Summary of GNN architectures used in our experiments. Each model is described by its layer composition, hidden units, activation functions, dropout strategy, and notable characteristics.

| Model | Layers | Hidden Units | Activation | Dropout | Learning rate | Decay | Epoch |
|---|---|---|---|---|---|---|---|
| GCN | 3 × GCNConv | 64 → 64 → Output | ReLU | Yes (intermediate layers) | 0.003 | 0.0005 | 500 |
| APPNP | Linear → Linear → APPNP | 64 → 64 → 10 → Output | ReLU | Yes (before Linear layers) | 0.003 | 0.0005 | 500 |
| GAT | 2 × GATv2Conv | 64 × 8 → Output | ELU | Yes (p=0.6) | 0.003 | 0.0005 | 500 |
| GIN | 2 × GATv2Conv | 64 × 8 → Output | ELU | Yes (p=0.6) | 0.003 | 0.0005 | 500 |
| GraphSAGE | 2 × SAGEConv | 64 → Output | ReLU | Yes (after first layer) | 0.003 | 0.0005 | 500 |

(2025), have demonstrated strong performance across a range of applications Li & Goldwasser (2019); Paliwal et al. (2019); Pfaff et al. (2020); Ying et al. (2018). Nevertheless, their scalability to large graphs remains a significant bottleneck. Motivated by recent efforts in scalable learning Huang et al.

(2021), we explore how our graph coarsening framework can improve the efficiency and scalability of GNN training, enabling more effective processing of large-scale graph data. Specifically, we train several GNN models on the coarsened version of the original graph while evaluating their performance on the original graph's test nodes. As discussed earlier in 4.4, our experimental setup spans a diverse collection of datasets, each with distinct structural characteristics. For *homophilic* graph settings, we follow the architectural configurations proposed in UGC Kataria et al. (2024), see Table 11. For *heterophilic* graphs, the GNN model designs are based on the implementations introduced in Lim et al. (2021). The *heterogeneous* GNN architectures are adopted directly from Gao et al. (2024a).

Table 12 reports node classification accuracy for homophilic and Table 13 reports node classification accuracy for heterophilic graphs. The AH-UGC framework consistently delivers results that are either on par with or exceed the performance of existing coarsening methods. As shown in Table 3, the framework is independent of any particular GNN architecture, highlighting its robustness and model-agnostic characteristics.

Table 12: Node classification accuracy (%) for homophilic datasets

| Dataset | Model | VAN | VAE | VAC | HE | aJC | aGS | Kron | UGC | AH-UGC | Base |
|---------|-------|-----|-----|-----|-----|-----|-----|------|-----|--------|------|
| DBLP | GCN | 79.65 | 80.36 | 80.55 | 79.99 | 80.55 | 79.26 | 79.40 | 85.75 | 80.27 | 84.00 |
| | SAGE | 80.58 | 80.07 | 80.16 | 80.81 | 80.61 | 81.57 | 79.48 | 68.56 | 68.31 | 84.08 |
| | GIN | 79.40 | 79.20 | 80.38 | 78.83 | 77.96 | 78.18 | 78.01 | 73.95 | 79.82 | 83.26 |
| | GAT | 74.43 | 78.32 | 76.49 | 77.56 | 78.97 | 77.51 | 75.93 | 77.93 | 79.48 | 82.25 |
| | APPNP | 84.25 | 83.80 | 83.63 | 83.60 | 83.29 | 84.25 | 84.05 | 84.84 | 85.18 | 85.75 |
| CS | GCN | 91.63 | 92.01 | 91.19 | 92.03 | 91.41 | 87.26 | 92.55 | 92.66 | 92.47 | 93.51 |
| | SAGE | 94.32 | 94.19 | 94.57 | 94.24 | 93.94 | 93.70 | 94.02 | 89.17 | 89.83 | 94.82 |
| | GIN | 89.80 | 89.69 | 89.83 | 90.70 | 89.61 | 88.00 | 90.64 | 86.77 | 81.07 | 83.50 |
| | GAT | 91.98 | 91.52 | 92.31 | 91.57 | 90.67 | 91.19 | 89.50 | 89.83 | 90.48 | 91.84 |
| Citeseer | GCN | 66.22 | 67.72 | 67.12 | 68.02 | 67.27 | 65.92 | 66.67 | 65.31 | 65.46 | 70.12 |
| | SAGE | 64.71 | 72.52 | 70.87 | 63.96 | 66.06 | 72.37 | 73.42 | 61.71 | 64.26 | 74.47 |
| | GIN | 68.17 | 69.82 | 68.77 | 70.57 | 69.70 | 67.87 | 68.02 | 64.41 | 63.66 | 71.62 |
| | GAT | 71.17 | 70.87 | 71.02 | 72.07 | 71.17 | 68.92 | 71.47 | 65.76 | 69.21 | 71.32 |
| | APPNP | 70.42 | 71.32 | 70.27 | 68.02 | 71.17 | 71.32 | 69.82 | 68.61 | 69.06 | 73.12 |
| PubMed | GCN | 85.73 | 86.74 | 86.66 | 87.60 | 86.11 | 86.08 | 86.11 | 84.66 | 85.47 | 87.60 |
| | SAGE | 87.40 | 86.11 | 87.15 | 66.45 | 86.49 | 87.45 | 87.73 | 87.34 | 72.16 | 88.28 |
| | GIN | 81.98 | 82.07 | 82.78 | 60.11 | 79.03 | 82.96 | 81.49 | 82.42 | 83.97 | 85.75 |
| | GAT | 84.32 | 69.78 | 81.11 | 50.60 | 75.99 | 84.23 | 83.90 | 84.66 | 84.63 | 87.39 |
| | APPNP | 86.89 | 87.20 | 88.21 | 87.70 | 87.12 | 86.84 | 87.22 | 85.64 | 85.80 | 87.88 |
| Physics | GCN | 94.75 | 94.62 | 94.57 | 94.73 | 94.39 | 94.75 | 94.40 | 95.20 | 94.88 | 95.79 |
| | SAGE | 96.26 | 96.04 | 96.08 | 95.97 | 96.04 | 96.18 | 96.01 | 95.21 | 95.78 | 96.44 |
| | GIN | 94.90 | 94.56 | 94.78 | 94.49 | 93.79 | 94.79 | 92.65 | 94.41 | 94.94 | 95.66 |
| | GAT | 94.97 | 95.01 | 95.00 | 94.65 | 95.36 | 94.60 | 94.85 | 96.02 | 95.10 | 94.28 |
| | APPNP | 96.20 | 96.20 | 96.28 | 96.11 | 95.97 | 96.07 | 96.21 | 96.17 | 96.10 | 96.28 |

## M.1 Transformer based models for node classification.

We further include node-classification accuracies of recent Graph Transformer models like Nodeformer Wu et al. (2022) and SGFormer Wu et al. (2023), see Table 14. We observed that coarsening methods consistently do fairly well for GNNs (GCN, GAT, etc.), but the gains are much smaller or even negative for transformer-based methods. Our current understanding is transformer-based methods rely more on global attention and fine-grained token distinctions that may be partly lost after node merging. Moreover, graph Transformers also depend on positional/structural encodings (e.g., distance/Laplacian features) that our coarsening is not specifically designed to preserve, and their larger capacity makes them more prone to overfitting or instability on aggressively coarsened graphs. We will add a short discussion of these points in the revised version and explicitly state that improving compatibility with Transformer-style GNNs is an interesting direction for future work.

Table 13: Node classification accuracy (%) for heterophilic datasets.

| Dataset | Model | VAN | VAE | VAC | HE | aJC | aGS | Kron | UGC | AH-UGC | Base |
|---|---|---|---|---|---|---|---|---|---|---|---|
| Film | SGC | 29.36 | 27.84 | 29.95 | 26.15 | 26.89 | 25.74 | 27.74 | 21.47 | 21.68 | 27.63 |
| | Mixhop | 28.21 | 30.68 | 29.84 | 29.52 | 29.10 | 29.15 | 31.15 | 21.57 | 21.79 | 30.92 |
| | GCN2 | 26.15 | 28.47 | 28.00 | 26.94 | 27.63 | 25.84 | 29.42 | 19.47 | 20.42 | 28.36 |
| | GPR-GNN | 26.52 | 27.95 | 27.10 | 27.74 | 26.78 | 28.36 | 28.26 | 20.68 | 21.31 | 29.73 |
| | GatJK | 26.11 | 25.89 | 25.79 | 25.10 | 25.31 | 25.31 | 26.63 | 22.42 | 21.21 | 23.94 |
| deezer-europe | SGC | 54.55 | 55.31 | 54.50 | 55.38 | 54.48 | 54.69 | 55.15 | 54.49 | 55.06 | 57.08 |
| | Mixhop | 58.42 | 59.10 | 58.48 | 58.82 | 58.34 | 57.38 | 58.80 | 59.78 | 60.98 | 64.31 |
| | GCN2 | 57.79 | 58.34 | 57.76 | 58.34 | 57.15 | 57.57 | 58.25 | 58.00 | 58.46 | 60.88 |
| | GPR-GNN | 56.30 | 56.85 | 56.70 | 56.77 | 55.73 | 55.55 | 56.31 | 58.44 | 58.46 | 56.97 |
| | GatJK | 55.21 | 57.50 | 54.63 | 55.76 | 55.31 | 56.03 | 56.87 | 57.01 | 57.33 | 59.01 |
| Amherst41 | SGC | 61.42 | 63.19 | 59.06 | 60.83 | 63.39 | 62.99 | 63.78 | 78.74 | 73.82 | 73.46 |
| | Mixhop | 59.25 | 58.46 | 57.68 | 58.66 | 59.06 | 63.78 | 58.66 | 69.29 | 64.37 | 72.48 |
| | GCN2 | 62.99 | 62.01 | 60.63 | 59.25 | 58.66 | 60.63 | 56.50 | 71.06 | 68.50 | 71.74 |
| | GPR-GNN | 59.45 | 58.86 | 58.07 | 55.91 | 57.68 | 59.25 | 55.71 | 66.73 | 63.98 | 60.93 |
| | GatJK | 57.48 | 63.58 | 60.24 | 62.99 | 61.61 | 64.76 | 62.60 | 64.37 | 67.72 | 78.13 |
| Johns Hopkins55 | SGC | 62.72 | 69.19 | 68.77 | 69.35 | 68.85 | 70.28 | 69.19 | 73.80 | 72.96 | 73.77 |
| | Mixhop | 63.64 | 65.74 | 68.18 | 64.90 | 62.22 | 64.90 | 63.73 | 69.94 | 67.25 | 73.56 |
| | GCN2 | 66.16 | 67.51 | 67.42 | 64.23 | 65.49 | 65.74 | 64.40 | 71.12 | 65.24 | 73.45 |
| | GPR-GNN | 62.05 | 63.06 | 62.30 | 62.80 | 60.37 | 61.96 | 61.71 | 66.33 | 63.31 | 64.95 |
| | GatJK | 62.80 | 69.10 | 67.34 | 66.41 | 65.99 | 65.58 | 67.00 | 69.77 | 65.32 | 77.12 |
| Reed98 | SGC | 53.46 | 57.14 | 53.92 | 52.07 | 55.30 | 58.06 | 53.92 | 57.60 | 57.60 | 68.79 |
| | Mixhop | 50.69 | 58.99 | 49.77 | 48.85 | 55.30 | 59.45 | 53.46 | 60.37 | 52.53 | 62.43 |
| | GCN2 | 56.68 | 59.45 | 51.61 | 50.69 | 51.61 | 56.68 | 50.69 | 61.75 | 57.14 | 64.16 |
| | GPR-GNN | 48.39 | 57.60 | 48.39 | 45.62 | 55.76 | 58.06 | 53.46 | 57.60 | 54.84 | 56.07 |
| | GatJK | 55.30 | 58.99 | 53.00 | 51.61 | 51.61 | 56.22 | 53.92 | 62.67 | 60.83 | 69.94 |
| Squirrel | SGC | 31.97 | 33.13 | 30.98 | 36.66 | 34.97 | 36.59 | 35.59 | 40.89 | 39.51 | 43.61 |
| | Mixhop | 36.28 | 30.21 | 24.60 | 34.90 | 28.44 | 27.90 | 37.05 | 46.12 | 43.97 | 46.40 |
| | GCN2 | 39.74 | 42.28 | 39.20 | 41.74 | 37.97 | 39.12 | 41.51 | 43.12 | 44.35 | 50.72 |
| | GPR-GNN | 29.36 | 25.67 | 28.82 | 28.82 | 26.44 | 27.06 | 30.59 | 45.12 | 43.74 | 34.39 |
| | GatJK | 31.44 | 37.43 | 32.82 | 46.12 | 38.36 | 37.89 | 46.81 | 40.89 | 39.43 | 46.01 |
| Chameleon | SGC | 38.60 | 51.58 | 45.79 | 54.91 | 52.63 | 53.15 | 54.39 | 58.60 | 59.65 | 57.46 |
| | Mixhop | 40.53 | 51.40 | 43.33 | 50.35 | 49.82 | 49.30 | 54.39 | 58.25 | 58.60 | 63.16 |
| | GCN2 | 47.37 | 52.11 | 56.84 | 59.30 | 59.65 | 58.95 | 59.12 | 51.40 | 49.82 | 67.11 |
| | GPR-GNN | 40.53 | 46.32 | 41.05 | 39.64 | 40.35 | 43.68 | 51.05 | 54.74 | 52.28 | 55.04 |
| | GatJK | 41.40 | 52.46 | 36.49 | 60.00 | 56.49 | 55.96 | 62.63 | 54.39 | 55.44 | 71.05 |
| Cornell | SGC | 67.24 | 67.09 | 68.26 | 68.02 | 68.35 | 69.02 | 68.33 | 76.68 | 76.08 | 72.78 |
| | Mixhop | 66.79 | 67.67 | 67.14 | 66.07 | 66.45 | 66.71 | 66.41 | 70.64 | 71.61 | 76.49 |
| | GCN2 | 66.31 | 66.83 | 66.98 | 67.64 | 67.17 | 62.91 | 66.50 | 72.71 | 70.90 | 77.18 |
| | GPR-GNN | 64.98 | 64.27 | 65.17 | 65.00 | 63.55 | 63.67 | 63.48 | 69.66 | 68.00 | 67.46 |
| | GatJK | 63.48 | 65.31 | 68.28 | 66.00 | 67.40 | 66.21 | 66.64 | 70.09 | 70.35 | 78.37 |
| Penn94 | SGC | 62.93 | 62.33 | 62.23 | 62.13 | 63.52 | 63.03 | 63.52 | 75.74 | 75.87 | 66.78 |
| | Mixhop | 71.71 | 69.62 | 69.35 | 68.36 | 67.98 | 68.40 | 67.98 | 73.36 | 72.13 | 80.28 |
| | GCN2 | 71.79 | 69.55 | 70.75 | 69.52 | 69.61 | 71.41 | 69.61 | 71.85 | 72.07 | 81.75 |
| | GPR-GNN | 68.18 | 68.19 | 68.36 | 68.20 | 67.77 | 68.15 | 68.11 | 67.93 | 68.55 | 79.43 |
| | GatJK | 67.94 | 67.05 | 66.73 | 66.21 | 66.34 | 66.06 | 66.33 | 69.23 | 69.26 | 80.74 |

Table 14: Node classification accuracy (%) for select datasets in transformer based GNNs.

| Dataset | Model | AH-UGC | UGC | VAN | VAE |
|---|---|---|---|---|---|
| DBLP | Nodeformer | 76.07 | 71.05 | 73.53 | 71.33 |
| | SGFormer | 72.74 | 68.25 | 79.59 | 74.21 |
| Physics | Nodeformer | 79.98 | 90.00 | 90.89 | 49.95 |
| | SGFormer | 92.18 | 93.65 | 94.97 | 94.00 |
| Squirrel | Nodeformer | 24.90 | 37.89 | 27.97 | 24.51 |
| | SGFormer | 31.20 | 43.65 | 37.66 | 31.43 |
| Chameleon | Nodeformer | 36.14 | 46.49 | 35.61 | 42.98 |
| | SGFormer | 47.36 | 49.29 | 47.30 | 50.17 |
| Cornell | Nodeformer | 65.95 | 57.44 | 19.14 | 70.21 |
| | SGFormer | 48.93 | 31.91 | 51.06 | 59.57 |

