# OpenReview forum: "AH-UGC: $\underline{\text{A}}$daptive and $\underline{\text{H}}$eterogeneous-$\underline{\text{U}}$niversal $\underline{\text{G}}$raph $\underline{\text{C}}$oarsening"
_ICLR.cc/2026/Conference — Submitted to ICLR 2026_

### Official Review · Reviewer_EQiL · 2025-10-27

**Soundness:** 2
**Presentation:** 2
**Contribution:** 3
**Rating:** 4
**Confidence:** 5

**Summary:**

This paper proposes "Adaptive Coarsening via Consistent and LSH Hashing" (AH-UGC), an extension of the previous work UGC. The method aims to provide a unified graph coarsening framework that supports adaptive, streaming, expanding, heterophilic, and heterogeneous graphs.

**Strengths:**

- The goal of creating a unified framework for diverse graph types (heterophilous, heterogeneous, streaming) and tasks (link prediction) is ambitious and highly relevant.
- The experimental evaluation is comprehensive, and the authors compare against up-to-date baselines.

**Weaknesses:**

1. The primary novelty appears to be the "consistency hash." The justification for this module is weak. Why does a balanced supernode formation necessarily benefit both GNN performance and spectral preservation? This claim is counter-intuitive and requires stronger theoretical guarantees or empirical analysis.
2. The method addresses heterophily by using a *global* edge heterophily factor during a *local* concatenation of $X$ and $A$. This design choice seems inconsistent. Is this approach theoretically sound? Please provide more ablation studies to justify this specific design and cite prior works [1][2] that use a similar metric or approach.
3. The "type-isolated coarsening" is a key component for handling heterogeneous graphs. To fairly assess its contribution, can this module also be applied to baselines like UGC? This would help clarify if the impurity reduction is due to the new module or other aspects of AH-UGC.
4. In Table 1, FGC and LAGC are reported as "OOT" (Out-of-Time) on PubMed, yet their original papers report results for this dataset. Please provide implementation details and analyze the bottleneck causing this issue. Is it due to a difference in hardware, setup, or implementation?
5. The paper notes that the method does not perform well with Transformer-based GNNs. Please provide a deeper analysis of why this limitation exists.
6. The paper fails to introduce or define all baseline methods. For instance, what are "aJC" and "aGS"?
7. The related work section is incomplete. Please include and discuss relevant works on:
    ◦ Streaming graph condensation [3][4]
    ◦ Heterogeneous graph condensation [5]
8. I deeply suggest the authors to narrow the scope and focus on one dimension. It's ambitious to propose a unified framework but it'd be better to do it in a dissertation. Putting every dimension in one conference paper requires the method to be very solid, self-contained and consistent with the tasks you want to solve. But this proposed method (especially the novel part) is only targeted for balanced coarsening, which is not related to heterophily, heterogeneousity or streaminig.

Minor:

- **Table 1:** Please right-align the numbers in Table 1 for better readability.
- **Figure 2:** The figure quality is low. Please use a vector format (e.g., PDF) and enlarge the font size.
- **Table 4 & 9:** Highlights for the best-performing methods are missing in Table 4 and for the "Cora" rows in Table 9.
- **Table 10:** The GCN model is set to 3 convolution layers, which is non-standard (typically 2). Please justify this choice.
- **Figure 4:** Please define "hDBLP" in the figure caption or text.

## References:

[1] Beyond Homophily in Graph Neural Networks: Current Limitations and Effective Designs, In NeurIPS 2020

[2] Large Scale Learning on Non-Homophilous Graphs: New Benchmarks and Strong Simple Methods, In NeurIPS 2021

[3] Scalable graph condensation with evolving capabilities, arxiv 2025

[4] Graph Condensation for Open-World Graph Learning, In KDD 2024

[5] Training-Free Heterogeneous Graph Condensation via Data Selection, In ICDE 2025

**Questions:**

See Weaknesses.

---

> ### Author Response · Authors · 2025-11-21
>
> We thank the reviewer for their valuable comments and insights and for taking the time to go through our paper.
>
> ### **Weaknesses**
>
> **1)** The primary novelty appears to be ..
>
> **Ans)** We would like to clarify the role of consistent hashing in AH-UGC. Beyond the consistency hash itself, the main novelty of AH-UGC is that it jointly addresses adaptive and heterogeneous graph coarsening: the introduction (lines 77–86) explicitly explains (i) the need for multiple coarsened graphs at different resolutions and (ii) the need to respect node/edge types in heterogeneous graphs, and how AH-UGC is designed around these two goals.
>
> The purpose of combining CH with LSH is primarily algorithmic: it lets us avoid the recursive bin-width search of UGC and efficiently generate an entire family of coarsened graphs from a single sorted LSH list, which existing methods cannot do without rerunning coarsening from scratch for each ratio.
>
> The “balanced supernodes” effect should therefore be understood as an intuitive side benefit, not as a strict theoretical guarantee that “necessarily” improves everything.
> Concretely:
> * Minority-class preservation. When the data are imbalanced, very large bins (as in fixed bin-width) can easily swamp minority nodes. CH’s tendency to form more size-balanced supernodes increases the chance that small classes still form their own supernodes instead of being absorbed into dominant ones.
> * In regions where hash scores are very dense or very sparse, fixed bin-width can create some extremely large and some tiny supernodes, leading to highly uneven degrees and potentially unstable spectra. CH mitigates this by avoiding such extreme size disparities, which empirically leads to more stable GNN performance and spectral behavior.
>
> **2)** The method addresses heterophily...
>
> **Ans)** In order to create a GC framework suitable for heterophilic graphs, it is important to consider features at both i) the node level, i.e., features, and ii) the structure-level, i.e., adjacency matrix, together. The augmented feature construction $F = (1 - \alpha) \cdot X \oplus \alpha \cdot A$
> (with a global heterophily factor $\alpha$) is inherited from UGC and AH-UGC follows the same design to handle heterophilic graphs.
>
> Intuitively, $\alpha$ acts as a global summary statistic that interpolates between purely feature-based ($\alpha = 0$) and strongly structure-informed ($\alpha = 1$) representations, which is consistent with prior work.
>
> To address the reviewer’s request, we now include an ablation study over $\alpha$ for two heterophilic datasets (Squirrel and Chameleon), where we sweep $\alpha$ from 0 to 1. The table shows that GCN performance peaks when $\alpha$ is set close to the empirical heterophily factor of the graph (near 0.78 for both datasets), supporting the soundness and usefulness of this global scaling.
>
> | $\alpha$ value | GCN accuracy on Squirrel | GCN accuracy on Chameleon |
> |-|-|-|
> | 0.0 | 20.39 | 30.27 |
> | 0.1 | 24.62 | 37.68 |
> | 0.2 | 26.35 | 43.29 |
> | 0.3 | 27.41 | 43.51 |
> | 0.4 | 27.89 | 42.63 |
> | 0.5 | 28.21 | 39.54 |
> | 0.6 | 28.44 | 47.38 |
> | 0.7 | 30.11 | **49.97** |
> | 0.8 | **31.18** | 48.96 |
> | 0.9 | 29.57 | 47.20 |
> | 1.0 | 28.92 | 45.72 |
>
> We have now added the missing references [1,2] and explicitly cited them in Section 3.1.
>
> **3)** The "type-isolated coarsening" is a key..
>
> **Ans)** Conceptually, the proposed type–isolated coarsening is an orthogonal module: it enforces that only nodes of the same type are merged and that the resulting superedges respect the original type schema. This constraint can indeed be plugged into any merge-based coarsening method, including UGC—for example, by (i) running UGC independently on each node type or (ii) restricting UGC’s merges so that cross-type merges are forbidden.
>
> In our experiments we deliberately kept all baselines unmodified to reflect their original design and reported implementations, and to avoid retrofitting them with additional type information they were not built to exploit.
>
> As suggested by the reviewer, we have conducted additional experiments to isolate this effect by comparing A-UGC with AH-UGC. Here, A-UGC uses the same pipeline as AH-UGC but without enforcing type isolation, while AH-UGC includes the type-isolated coarsening module; the results of this comparison are reported below.
>
> ### Comparison of A-UGC (without type isolation) and AH-UGC (with type-isolated coarsening), showing that enforcing type isolation consistently improves accuracy on heterogeneous graphs.
> | Dataset| Model | A-UGC   | AH-UGC |
> |--|--|--|--|
> | IMDB |  HeteroSGC| 48.91 | 51.46  |
> |   | HeteroGCN | 47.14 | 52.91  |
> |   | HeteroGCN2 | 45.93 | 52.58  |
> | hDBLP |  HeteroSGC| 50.82 | 56.60  |
> |    | HeteroGCN | 57.32 | 63.13  |
> |   | HeteroGCN2  | 40.28 | 62.71  |
> | ACM  |  HeteroSGC | 51.13 | 53.82  |
> |    | HeteroGCN   | 40.47 | 85.16  |
> |  | HeteroGCN2  | 49.02 | 84.36  |

---

> ### Author Response · Authors · 2025-11-21
>
> **4)** In Table 1, FGC and LAGC are reported as "OOT"..
>
> **Ans)** In our setting, we evaluate adaptive graph coarsening, where we need multiple coarsened graphs per dataset (10 different coarsening ratios in Table 1). This requires running each coarsening method repeatedly to obtain all resolutions. Under this multi-run setting, both FGC and LAGC took more than 86,400 seconds (> 1 day) to finish all required runs, which we mark as OOT (Out-of-Time). Importantly, this is not due to a change in hardware or an unfair setup, but to the computational profile of FGC/LAGC: they involve costly steps and must be re-executed from scratch for each target ratio. All methods, including ours and the baselines, were run on the same machine; we report full system specifications in Appendix A and repeat them here for completeness.
>
> **System Specifications:** All experiments are conducted on a server equipped with two NVIDIA RTX A6000 GPUs (48 GB memory each) and an Intel Xeon Platinum 8360Y CPU with 1 TB RAM.
>
> **5)** The paper notes that the method does not perform..
>
> **Ans)** We thank the reviewer for pointing this out and agree that this limitation deserves a clearer discussion. In our experiments we observed that coarsening methods consistently do fairly well for GNNs (GCN, GAT, etc.), but the gains are much smaller or even negative for transformer-based methods. Our current understanding is transformer-based methods rely more on global attention and fine-grained token distinctions that may be partly lost after node merging. Moreover, graph Transformers also depend on positional/structural encodings (e.g., distance/Laplacian features) that our coarsening is not specifically designed to preserve, and their larger capacity makes them more prone to overfitting or instability on aggressively coarsened graphs.
>
> We have added a short discussion of these points in the revised version and explicitly state that improving compatibility with Transformer-style GNNs is an interesting direction for future work.
>
> **6)** The paper fails to introduce or define ...
>
> **Ans)** We thank the reviewer for pointing this out. We now explicitly introduce and define aJC and aGS in the related work. We have also added a dedicated notation/abbreviation table to improve clarity and readability; for details on the notation, we kindly refer the reviewer to our response to **Reviewer rpLH, Q1**.
>
> **7)** The related work section is incomplete..
>
> **Ans)** We thank the reviewer for pointing out these missing connections. We have now incorporated and discussed the recent streaming graph condensation works [3,4,5] in the related work section, clarifying how they differ from and complement our setting. The corresponding additions and comparisons are highlighted in blue in the revised manuscript.
>
> **8)** I deeply suggest the authors ...
>
> **Ans)** We thank the reviewer for this thoughtful comment. First, we would like to clarify that balanced coarsening is a side result, not the primary objective, of our design. The core goals of AH-UGC are adaptive graph coarsening and heterogeneous graph coarsening; consistent hashing is introduced to make adaptive multi-resolution coarsening feasible from a single LSH ordering, and type-isolated coarsening ensures that heterogeneous node/edge types are respected. The “balanced supernodes’’ property emerges from this mechanism rather than being the main target of the method. For details on the novelty and the role of balanced supernodes, we kindly refer the reviewer to our response to Q1.
>
> Our intention was indeed to move toward a truly universal coarsening framework, and since AH-UGC is built on UGC, it naturally inherits UGC’s generality while adding two concrete, new capabilities: (a) adaptive coarsening (multiple resolutions from one run) and (b) heterogeneous coarsening (type isolation). The paper is structured to reflect exactly this focus: in the introduction, lines 77–86 explicitly formulate the two open problems (adaptive and heterogeneous); lines 141–149 state Goal 1 and Goal 2 around these; Section 3.1 and Section 3.2 then present the algorithms that realize each goal; and the experiments are organized to quantify both dimensions.
>
> We appreciate the reviewer’s concern that proposing a unified framework in a conference paper is ambitious, and we are grateful that this ambition is recognized. At the same time, we believe that the two added dimensions (adaptive + heterogeneous) are tightly coupled and implemented within a single, coherent extension of UGC.
>
> ### **Minor**
>
> a) The numbers in Table 1 have now been right-aligned.
>
> b)  We have regenerated Figure 2 in vector format and enlarged all font sizes.
>
> c)  We have now added highlights for the best-performing methods in Table 4 and for the Cora rows in Table 9.
>
> d)  In our experiments, setting the GCN depth to 3 layers was simply an arbitrary design choice rather than a carefully tuned hyperparameter.
>
> e)  We have now defined “hDBLP” in the caption of Figure 4.

---

> ### Author Response · Authors · 2025-11-24
>
> Dear Reviewer EQiL,
>
> Thank you again for taking the time to review our submission and for sharing your thoughtful feedback; it has been invaluable in helping us refine our work. We’ve done our best to address all the raised concerns carefully in the rebuttal, and **we’re very open to further discussion.**
>
> Since we’re midway through the discussion phase, we just wanted to gently check in to see if there are any remaining questions or points of clarification we could help with while there’s still time. **We’d be more than happy to elaborate on any aspect that might need further explanation.**
>
> And if you feel that your concerns have been adequately addressed, **we would be sincerely grateful if you would consider updating your rating to reflect your current assessment of the work.**
>
> Best,
>
> Authors

---

> > ### Author Response · Authors · 2025-11-27
> > **Appeal to Reviewer**
> >
> > Dear Reviewer,
> >
> > Thank you again for taking the time to review our submission and for sharing your thoughtful feedback. As we approach the end of the discussion phase, we wanted to gently follow up to see if there are any remaining concerns that we can help address. We’re more than happy to elaborate on any aspect that may still be unclear.
> >
> > **Your feedback is indeed important for determining the ultimate fate of our work.** If you feel that your concerns have been sufficiently resolved, **we would be sincerely grateful if you might consider revisiting your score to reflect your updated assessment.**
> >
> > Best,
> >
> > Authors

---

### Official Review · Reviewer_M9ms · 2025-10-29

**Soundness:** 3
**Presentation:** 3
**Contribution:** 3
**Rating:** 4
**Confidence:** 4

**Summary:**

This paper proposes AH-UGC, a unified framework for adaptive and heterogeneous graph coarsening. It integrates Locality-Sensitive Hashing (LSH) and Consistent Hashing (CH) to enable fast, scalable, and ratio-adaptive coarsening, while ensuring type-isolated merging for heterogeneous graphs. Experiments on 23 datasets show superior scalability and strong structural and downstream performance.

**Strengths:**

1. Novel adaptive mechanism combining LSH and CH for multi-resolution coarsening.
2. Type-consistent design effectively supports heterogeneous graphs.
3. Comprehensive experiments demonstrate scalability and robustness across diverse settings.

**Weaknesses:**

1. The paper does not include or discuss several recent graph coarsening or condensation methods from 2024–2025, which limits the completeness and fairness of the comparison.
2. The claim around line 177 that graph condensation methods are inefficient is not accurate. Recent studies such as [1], [2], and many more, have proposed efficient condensation techniques, and the paper would benefit from acknowledging and comparing to these works.

[1] Rethinking and Accelerating Graph Condensation: A Training-Free Approach with Class Partition, WWW 2025

[2] Adapting Precomputed Features for Efficient Graph Condensation, ICML 2025

**Questions:**

See weaknesses.

---

> ### Author Response · Authors · 2025-11-23
>
> We thank the reviewer for their valuable comments and insights and for taking the time to go through our paper.
>
> ### **Weaknesses**
>
> **1)** The paper does not include or discuss several recent graph coarsening or condensation methods from 2024–2025, which limits the completeness and fairness of the comparison.
>
> **Ans)** We thank the reviewer for pointing out these missing connections. We have now incorporated and discussed the recent works in the related work section, clarifying how they differ from and complement our setting. We have now added the missing references [1,2] and explicitly cited them in related works. The corresponding additions and comparisons are highlighted in blue in the revised manuscript.
>
> **2)** The claim around line 177 that graph condensation methods are inefficient is not accurate. Recent studies such as [1], [2], and many more, have proposed efficient condensation techniques, and the paper would benefit from acknowledging and comparing to these works.
>
> **Ans)** We thank the reviewer for this comment and agree that our original wording was too strong. Our intention was not to claim that all recent graph condensation methods are universally inefficient, but rather that they are model-dependent and training-dependent, which makes them less suitable as a generic coarsening primitive in the setting we study.
>
> Concretely, gradient–matching based condensation methods require training a GNN on the original large graph and repeatedly matching gradients between the real and synthetic graphs. This tightly couples them to a particular backbone, requires full supervision, and entails substantial training costs on the full graph before a condensed graph is obtained. As a result, while such methods can be very valuable for tasks like storage reduction, visualization, or speeding up a given supervised model, they are less appropriate as a lightweight, model-agnostic, and easily reusable coarsening operator.
>
> We have revised lines 174–177 to make this clearer, and we have also explicitly acknowledged recent efficient condensation works [1,2] in this discussion, for example, as
>
> "More recent streaming graph condensation methods GECC and OpenGC extend this line of work to evolving or open-world settings by updating the condensed graph as new data arrives, but they still operate in a supervised, model-specific regime and do not provide a generic, model-agnostic method. HGCond is  designed for heterogeneous graphs, yet it inherits these training and model-dependence limitations and does not support adaptive condensation. Recent works such as CGC[1], GCPA[2], OpenGC, and FreeHGC propose training-free graph condensation methods. However, CGC, GCPA and OpenGC do not support heterogeneous graphs, and none of these methods are adaptive in nature. Moreover, condensation-based approaches generate synthetic condensed datasets, whereas coarsening operates directly on the original graph, merging nodes while learning representations from the original data."
>
> In addition, as suggested we ran experiments with recent condensation methods CGC and GCPA and report their condensation times (in seconds) below:
>
> #### Total time (in seconds) to generate coarsened graphs at multiple resolutions, targeting a set of coarsening ratios of R = {0.55, 0.50, 0.45, 0.40, 0.35, 0.30, 0.25, 0.20, 0.15, 0.10}
> | Dataset | CGC   | GCPA  |
> |---------|-------|-------|
> | Cora    | 1.73  | 164   |
> | PubMed  | 1.86  | 266   |
> | Flickr  | 42283 | OOT |
> | Reddit  | OOT | OOT |
> | Arxiv | OOT | OOT |
>
> These results illustrate the computational phenomenon we referred to: condensation generates a synthetic graph whose size controls the number of learnable parameters. As the required synthetic graph becomes larger (e.g., for Flickr, Arxiv and Reddit), the number of parameters and the optimization cost explode, leading to very large condensation times. This makes such methods attractive for scenarios like deploying a fixed smaller model or storing a compact dataset representation, but it also highlights their computational drawbacks when used as a generic, repeatedly-invoked coarsening primitive in large-scale, adaptive settings—precisely the use case AH-UGC targets.

---

> ### Author Response · Authors · 2025-11-26
>
> Dear Reviewer M9ms,
>
> Thank you again for taking the time to review our submission and for sharing your thoughtful feedback; it has been invaluable in helping us refine our work. We’ve done our best to address all the raised concerns carefully in the rebuttal, and **we’re very open to further discussion.**
>
> Since we are near the end of the discussion phase, we just wanted to gently check in to see if there are any remaining questions or points of clarification we could help with while there’s still time. **We’d be more than happy to elaborate on any aspect that might need further explanation.**
>
> And if you feel that your concerns have been adequately addressed, **we would be sincerely grateful if you would consider updating your rating to reflect your current assessment of the work.**
>
> Best,
>
> Authors

---

> > ### Author Response · Authors · 2025-11-27
> > **Appeal to Reviewer**
> >
> > Dear Reviewer,
> >
> > Thank you again for taking the time to review our submission and for sharing your thoughtful feedback. As we approach the end of the discussion phase, we wanted to gently follow up to see if there are any remaining concerns that we can help address. We’re more than happy to elaborate on any aspect that may still be unclear.
> >
> > **Your feedback is indeed important for determining the ultimate fate of our work.** If you feel that your concerns have been sufficiently resolved, **we would be sincerely grateful if you might consider revisiting your score to reflect your updated assessment.**
> >
> > Best,
> >
> > Authors

---

### Official Review · Reviewer_jTzn · 2025-11-01

**Soundness:** 2
**Presentation:** 4
**Contribution:** 2
**Rating:** 6
**Confidence:** 3

**Summary:**

As large-scale data core sets and sketching gained significant attention some time ago, graph coarsening has become a crucial challenge in computer science for increasingly massive graphs. This paper addresses two research challenges as the two promised advances for recent breakthroughs: (1) achieving consistent and simultaneous coarsening across multiple resolutions, and (2) ensuring applicability even in scenarios where input and observed graphs are heterogeneous.

**Strengths:**

- The two problem settings in this paper (i.e., simultaneously achieving multiple levels of coarsening and handling heterogeneous data) appear to be both highly significant and novel within the context of graph coarsening. And very happily, the authors' method provides a framework that can unify and solve these two challenges simultaneously.

- The experimental investigations in this paper are exceptionally large-scale and comprehensive. As they are shared with the community alongside open-source code, they provide crucial assistance for subsequent research.

- The presentation of this paper is highly clear and polished. (However, I do feel there is a little room for improvement regarding the mathematical notation, and I have commented on this briefly in the questions section.)

**Weaknesses:**

- This paper represents a solid advancement of existing research [Kataria+, NeurIPS2024]. However, its improvements (in my personal, subjective view) may be relatively minor compared to the breakthroughs in the foundational research [Kataria+, NeurIPS2024] itself. Specifically, the paper certainly adds two functionalities: (1) simultaneously acquiring coarsening at multiple hierarchical levels, and (2) handling heterogeneous graphs. However, these additions appear to be straightforward extensions of existing work, and I personally do not perceive them as constituting the ‘an innovative and flexible approach’ claimed by the authors in line 192.


- I have remaining concerns regarding the novelty of the theoretical contributions of the proposed method discussed in Section 3.2 of this paper. I remain skeptical about the novelty of the theoretical contributions of the proposed method discussed in Section 3.2 of this paper. Theorem 3.1 and Lemma 1 appear to state general properties of Gaussian variables within a rather broad context. In fact, the h(x) used in these seems to be employed in a different sense than the score h used by the authors in lines 215 and elsewhere. While the facts stated in Section 3.3 suggest the (high-probability) correctness of the authors' AH-UGC behavior, I question whether these theoretical analyses themselves constitute the authors' contribution. I find the positioning of these theoretical analyses within the paper somewhat unclear.

**Questions:**

I think there is some room for improvement in the handling of notation. I will give one example here, but reconsidering the overall notation for similar reasons would likely enhance clarity of this excellent paper.

For example, line 198 first mentions the augmented vector F_{i}. Later, on line 202, the expression ‘Let F_{I}’ appears. As a reader, I would greatly appreciate it if a symbol were properly defined upon its first appearance and then consistently reused within the same context. Considering this, I personally find the following style easier to read:

- It would be helpful if line 198 explicitly stated that it provides the definition of F_{i}. For example, a simple expression like ‘F_{I} :=‘ might suffice.

- Line 202 could be made much clearer by emphasizing that it is merely reintroducing a variable already defined once before. For instance, writing something like ‘Recall that F_{i} is defined at Line 198’ would greatly improve readability.

Is the equation in line 206 correctly written? I believe that with this notation, the right-hand side does not appear to be a scalar. More precisely, is it W_{k}^{\top}\cdot F_{I}+b_{k}?

Since the heading in Section 3.1 is labeled ‘Goal1,’ to maintain consistency in this notation, it would be clearer if ‘Goal2’ were also explicitly stated. Specifically, line 239 would be ‘Goal2,’ correct?

Is the equation on line 266 a period (.) rather than a comma (,)?

---

> ### Author Response · Authors · 2025-11-21
>
> We thank the reviewer for their valuable comments and insights and for taking the time to go through our paper.
>
> ### **Weaknesses**
>
> **1)** This paper represents a solid advancement of existing research..
>
> **Ans)** We would like to clarify about AH-UGC novelty. While AH-UGC builds on UGC’s core idea of LSH-based coarsening, it introduces two key innovations that substantially extend its capabilities:
>
> * Unlike UGC, which requires recomputing the entire coarsening process for each ratio (via bin-width tuning), AH-UGC introduces consistent hashing that enables multi-resolution coarsening in a single pass, with theoretical guarantees (Theorem 3.1, Lemma 1). This makes AH-UGC fundamentally adaptive and incrementally refinable — a feature UGC lacks.
> * AH-UGC is the first to coarsen heterogeneous graphs while preserving type semantics. We introduce type-isolated coarsening and per-type compression strategies (Section 3.2), allowing us to handle varying feature dimensions and avoid type mixing — a critical limitation in UGC.
>
> We also want to clarify that, empirically, AH-UGC is 4–5× faster than UGC when generating multiple coarsened graphs (Table 1), and achieves +20–30% absolute gain in classification accuracy on heterogeneous datasets (Table 4), demonstrating clear practical advantages.
>
> **2)** I have remaining concerns regarding the novelty of the theoretical contributions..
>
> **Ans)** We thank the reviewer for raising this point and for pushing us to clarify the role and positioning of Section 3.2.
>
> Our intention with Theorem 3.1 and Lemma 1 is not to claim a deep new result in probability, but to justify why the LSH-based scoring we use in AH-UGC behaves as intended: if two nodes are similar in feature space, then, under a Gaussian random projection,
>
> * Their projected values h(x) will be close with high probability, and
> * Hence they are likely to receive similar hash scores and be placed close to each other in the global ordering that drives coarsening.
> In that sense, these results formalize the LSH consistency we rely on: the projection matrix is sampled from a Gaussian (stable) distribution, and the statements in Theorem 3.1 / Lemma 1 are used to argue that similar nodes remain close after projection and hashing.
>
> We agree that these are general properties of Gaussian variables; our contribution here is to specialize and connect them explicitly to AH-UGC’s coarsening mechanism rather than to present them as standalone theoretical novelties. If the reviewer prefer, we are happy to relabel Theorem 3.1 as a lemma and move the detailed statement to the appendix.

---

> ### Author Response · Authors · 2025-11-21
>
> ### **Questions**
>
> **1)** I think there is some room for improvement in the handling of notation....
>
> **Ans)** We thank the reviewer for highlighting the need for clearer and more consistent notation. Following this suggestion, we have modified Line 202 and included the $F_{i}$ definition in line 198. We have also added a dedicated notation table (included below) in Appendix A and explicitly refer to it in Section 2 so that readers can quickly look up symbols.
>
> | Symbols and abbreviations | Description |
> |-|-|
> | GC| Graph coarsening |
> | AH-UGC| Adaptive heterogeneous universal graph coarsening |
> | GNN| Graph neural network |
> | HGNN| Heterogeneous graph neural network |
> | LSH| Locality-Sensitive Hashing |
> | CH| Consistent Hashing |
> |UGC | Universal Graph Coarsening |
> |VAN |  Local Variation Neighbourhood |
> |VAE |  Local Variation Edge |
> |VAC |  Local Variation Clique |
> |HE | Heavy Edge Matching|
> |aJC | Algebraic Distance|
> |aGS |  Affinity GS|
> |FGC | Featured Graph Coarsening|
> | $\mathcal{G}$ | Graph |
> | $V$  | Set of vertices |
> | $E$ | Set of edges |
> | $A$ | Adjacency matrix of $G$ |
> | $X$  | Node feature matrix |
> | $D$  | Degree matrix |
> | $L$ | Graph Laplacian matrix|
> | $\mathcal{G}(V, A, X)$   | Original graph with vertices, adjacency, and features |
> | $\mathcal{G}(L, X)$   | Original graph with laplacian and features |
> | $N$ | Number of nodes |
> | $m$ | Number of edges |
> | $\Phi$| Node-type map|
> | $\Psi$ | Edge-type map|
> | $T_V$ | Set of node types |
> | $T_E$| Set of edge/relationship types |
> | $\mathcal{G}(V, E, \Phi, \Psi)$| Entity based Heterogeneous Graph|
> | $\mathcal{G}(X_{node\_type},A_{edge\_type}, y_{target\_type})$|Type based Heterogeneous Graph|
> | $r$| Coarsening ratio (fraction of nodes kept) |
> | $\mathcal{C}$ | Coarsening matrix (nodes $\rightarrow$ supernodes) |
> | $\mathcal{C}^{(r)}$ | Coarsening matrix at ratio $r$ |
> | $\mathcal{G}_c$ | Coarsened graph |
> | $\tilde V$ | Set of supernodes |
> | $A_c$| Coarsened adjacency matrix |
> | $X_c$  | Coarsened node feature matrix |
> | $\alpha$|Heterophily factory|
> | $\mathcal{W}$|Projection matrix|
> | $F$|Augumented Feature matrix|
> | $\pi$|Node to supernode mapping|
> | $h_i$ | Hash score for node i derived from features |
> |$\mathcal{R}$|List of coarsening ratios|
> |$\mathcal{L}^{t}$|Ordered list of hash values at timestamp t|
> |$\mathcal{C}_{\mathcal{H}}$|Set of coarsening matrices for heterogeneous graph|
>
> We thank the reviewer for catching other typos and fully agree with the comment.
>
> * The equation in line 206 is indeed corrected to $W_{k}^{\top}\cdot F_{i}+b_{k}$
> * We now explicitly restate Goal 2 at the point where we introduce heterogeneous graph coarsening (Section 3.2)
> * We have also added the missing period in line 266.

---

> ### Author Response · Authors · 2025-11-24
>
> Dear Reviewer jTzn,
>
> Thank you again for your thoughtful and positive review of our submission, we truly appreciate your assessment and the time you invested in reading and evaluating our work. We have carefully addressed all of your comments in the rebuttal and revised draft, and **we’re very open to further discussion.**
>
> Since we’re midway through the discussion phase, we just wanted to gently check in to see if there are any remaining questions or points of clarification we could help with while there’s still time. **We’d be more than happy to elaborate on any aspect that might need further explanation.**
>
> And if you feel that your concerns have been adequately addressed, **we would be sincerely grateful if you would consider updating your rating to reflect your current assessment of the work.**
>
> Thank you once again for your constructive feedback and support.
>
> Best regards,
>
> Authors

---

> > ### Comment · Reviewer_jTzn · 2025-11-28
> >
> > I am very grateful for the authors' thoughtful responses. I will carefully read through the responses and the revised paper this weekend. I apologize for causing the authors concern by not responding immediately. I ask for a little more time.

---

### Official Review · Reviewer_rpLH · 2025-11-02

**Soundness:** 2
**Presentation:** 1
**Contribution:** 3
**Rating:** 4
**Confidence:** 3

**Summary:**

This paper introduces a unified graph coarsening framework that simultaneously supports adaptive coarsening ratio and heterogeneous graph structures.
The method combines Locally Sensitive Hashing (LSH) with Consistency Hashing (CH) to rapidly generate multi-level coarsened graphs without recomputation, while preserving semantic consistency in heterogeneous graphs through a type-isolation strategy.
Experiments demonstrate that AH-UGC outperforms existing methods in efficiency, structural preservation, and downstream task performance.

**Strengths:**

This paper addresses two issues crucial to graph coarsening: adaptive coarsening ratios and heterogeneous graphs. The experimental results suggest that this method appears to be quite effective. The boundaries provided by the two theorems are quite interesting.

**Weaknesses:**

1. The writing of this paper need to be improved.It took me a long time to figure out what the author was writing about. After reading the paper multiple times, I still don't understand what the author is trying to convey in Goal $2$, and I couldn't find the definitions for some notations at all. Notation in this paper appears poorly organized; for example, matrices are represented using three distinct forms: uppercase letters, bold uppercase letters, and calligraphic letters. $r$ has multiple meanings in this paper (the r in Theorem 3.1 is defined differently from $r$ elsewhere). Given the abundance of symbols in this paper, I recommend the author create a notation table for readers to consult quickly. Another small suggestion is to use $\top$ (\top) instead of $T$ as the symbol for matrix transposition.

2. Apart from writing, this paper is a bit too incremental in terms of innovation relative to UGC. Especially when it comes to handling heterogeneous graphs, the overall approach is not novel for me.

3. The authors state that their method can be applied in streaming settings, but no relevant experiments have been conducted to validate this claim.

I like the methodology of this paper and I think that if the paper can improve their writing, the technical contribution of the paper should be a borderline accept.

**Questions:**

1. What is Goal $2$ trying to convey?
2. What is the time complexity of AH-UGC?
3. Is $l$ a hyperparameter? If so, how is it determined?

---

> ### Author Response · Authors · 2025-11-20
>
> We thank the reviewer for their valuable comments and insights and for taking the time to go through our paper.
>
> ### **Weaknesses**
>
> **1)** The writing of this paper need to be improved.It took me a long time to figure out what the author was writing about. After reading the paper multiple times, ...
>
> **Ans)** We thank the reviewer for this comment and we clarify the intended flow of the introduction. The introduction first recalls graph coarsening as a tool to compress large graphs, then discusses general limitations of existing coarsening methods, and finally narrows down to two concrete remaining challenges: (i) adaptive coarsening—obtaining multiple resolutions without recomputing from scratch, and (ii) heterogeneous coarsening—coarsening graphs with multiple node/edge types without breaking their semantic structure. Lines 077–086 explicitly describe these two challenges and provide example scenarios for each.
>
>
> **Regarding Goal 2**
>
>  We then introduce Goal 1 and Goal 2 precisely to address these two points: Goal 1 tackles the adaptive coarsening challenge, and Goal 2 tackles heterogeneous coarsening. In plain language, Goal 2 requires that when coarsening a heterogeneous graph, (a) nodes of different types are never merged into the same supernode, and (b) aggregated edges between supernodes preserve well-defined relation types and the original type-level schema, so that heterogeneous GNNs remain meaningful on the coarsened graph.
>
>  We agree that the formal constraints for Goal 2 are difficult to parse on first read. In the revision, we will move a short, plain-language explanation of both Goal 1 and Goal 2 before the formal equations, and explicitly point back to the motivating examples in lines 077–086 to make this connection clearer.
>
>  **Regarding Notations**
>
> We thank the reviewer for highlighting the need for clearer and more consistent notation. Following this suggestion, we have standardized matrix and graph symbols, removed unnecessary bold uppercase matrices, and ensured that each symbol has a single, consistent meaning throughout the paper. In particular, the symbol
> r is now used only as the coarsening ratio and is no longer used in Theorem 3.1, and we uniformly adopt the standard transpose notation $(\cdot)^\top$ instead of T. Following the reviewer’s suggestion, we have also added a dedicated notation table (included below) in Appendix A and explicitly refer to it in Section 2 so that readers can quickly look up symbols.
>
> | Symbols and abbreviations | Description |
> |-|-|
> | GC        | Graph coarsening |
> | AH-UGC    | Adaptive heterogeneous universal graph coarsening |
> | GNN       | Graph neural network |
> | HGNN      | Heterogeneous graph neural network |
> | LSH       | Locality-Sensitive Hashing |
> | CH        | Consistent Hashing |
> |UGC | Universal Graph Coarsening |
> |VAN |  Local Variation Neighbourhood |
> |VAE |  Local Variation Edge |
> |VAC |  Local Variation Clique |
> |HE | Heavy Edge Matching|
> |aJC | Algebraic Distance|
> |aGS |  Affinity GS|
> |FGC | Featured Graph Coarsening|
> |-|-|
> | $\mathcal{G}$ | Graph |
> | $V$            | Set of vertices |
> | $E$            | Set of edges |
> | $A$            | Adjacency matrix of $G$ |
> | $X$            | Node feature matrix |
> | $D$            | Degree matrix |
> | $L$            | Graph Laplacian matrix|
> | $\mathcal{G}(V, A, X)$   | Original graph with vertices, adjacency, and features |
> | $\mathcal{G}(L, X)$   | Original graph with laplacian and features |
> | $N$            | Number of nodes |
> | $m$            | Number of edges |
> | $\Phi$         | Node-type map |
> | $\Psi$         | Edge-type map |
> | $T_V$          | Set of node types |
> | $T_E$          | Set of edge/relationship types |
> | $\mathcal{G}(V, E, \Phi, \Psi)$| Entity based Heterogeneous Graph|
> | $\mathcal{G}(X_{node\_type},A_{edge\_type}, y_{target\_type})$|Type based Heterogeneous Graph|
> | $r$            | Coarsening ratio  |
> | $\mathcal{C}$            | Coarsening matrix  |
> | $\mathcal{C}^{(r)}$      | Coarsening matrix at ratio $r$ |
> | $\mathcal{G}_c$          | Coarsened graph |
> | $V_c$     | Set of supernodes |
> | $A_c$          | Coarsened adjacency matrix |
> | $X_c$          | Coarsened node feature matrix |
> | $\alpha$|Heterophily factory|
> | $\mathcal{W}$|Projection matrix|
> | $F$|Augumented Feature matrix|
> | $\pi$|Node to supernode mapping|
> | $h_i$ | Hash score for node i derived from features |
> |$\mathcal{R}$|List of coarsening ratios|
> |$\mathcal{L}^{t}$|Ordered list of hash values at timestamp t|
> |$\mathcal{C}_{\mathcal{H}}$|Set of coarsening matrices for heterogeneous graph|

---

> ### Author Response · Authors · 2025-11-20
>
> **2)** Apart from writing, this paper is a bit too incremental in terms of ...
>
>  **Ans)** While AH-UGC builds on UGC’s core idea of LSH-based coarsening, it introduces two key innovations that substantially extend its capabilities:
>
> * Unlike UGC, which requires recomputing the entire coarsening process for each ratio (via bin-width tuning), AH-UGC introduces consistent hashing that enables multi-resolution coarsening in a single pass, with theoretical guarantees (Theorem 3.1, Lemma 1). This makes AH-UGC fundamentally adaptive and incrementally refinable — a feature UGC lacks.
> * AH-UGC is the first to coarsen heterogeneous graphs while preserving type semantics. We introduce type-isolated coarsening and per-type compression strategies (Section 3.2), allowing us to handle varying feature dimensions and avoid type mixing — a critical limitation in UGC.
>
> We also want to clarify that, empirically, AH-UGC is 4–5× faster than UGC when generating multiple coarsened graphs (Table 1), and achieves +20–30% absolute gain in classification accuracy on heterogeneous datasets (Table 4), demonstrating clear practical advantages.
>
>  **3)** The authors state that their method can be applied in streaming settings, but no relevant experiments have been conducted to validate this claim.
>
>  **Ans)** Thank you for the suggestion. While our main focus in AH-UGC is on adaptivity and heterogeneous graphs, we agree that supporting streaming graphs is equally important. Since AH-UGC builds upon the principles of UGC (which is designed for streaming), our framework naturally supports incremental coarsening without recomputation.
>
> To further support this claim, we have now conducted additional experiments simulating streaming graph settings. Below are the accuracy and coarsening time results across time intervals for 3 datasets:
>
> ### Table: Accuracy and Coarsening Time for Streaming Graphs
>
> | Interval (% of data) | Cora (Acc — Time) | Pubmed (Acc — Time) | Physics (Acc — Time) |
> |-|-|-|-|
> |0–20|65.01-0.21|81.74-0.35|93.67-4.82|
> |20–30|73.43-0.06|82.91-0.09|94.17-0.99|
> |30–40|74.40-0.05|83.24-0.17|93.73-0.94|
> |40–50|80.01-0.09|83.98-0.21|94.59-1.10|
> |50–60|80.60-0.12|84.31-0.24|94.75-1.33|
> |60–70|82.69-0.13|84.51-0.30|94.90-1.37|
> |70–80|86.37-0.15|85.12-0.34|95.31-1.57|
> |80–90|86.00-0.18|85.32-0.35|96.40-1.39|
> |90–100|86.19-0.10|85.49-0.38|96.12-2.17|
>
> This table presents the **accuracy (Acc)** and **coarsening time (Time)** of AH-UGC on 3 benchmark datasets under a **streaming graph setting**. We assume data arrives in **incremental batches** (10–20% of nodes at a time), and coarsening is applied at each time step without recomputing the entire graph.
>
>  ### **Questions**
>
>  **1)** What is Goal 2 trying to convey?
>
>  **Ans)** Please see Weakness 1.
>
>  **2)** What is the time complexity of AH-UGC?
>
>  **Ans)** We thank the reviewer for this comment; below we provide a detailed time–complexity analysis, and we have also added the same discussion to Appendix I.
>
>  For **UGC**, the original paper shows that one run of the algorithm (feature projection + binning + building the coarsened adjacency) costs $\mathcal{O}(NLd + m),$ with $N = |V|$, $m = |E|$, $d$ the feature dimension, and $L$ the number of projectors.
>
> However, in the **adaptive** setting we need to find a suitable bin-width for each target coarsening ratio via the recursive *Bin-width Finder* algorithm of UGC. If this procedure is called $K$ times on average, then cost per ratio becomes $O\big((1+K)(NLd + m)\big).$ If we require $|R|$ different coarsening ratios, the overall cost is
> $$T_{UGC, adaptive} = \mathcal{O}\big(|R|(1+K)(NLd + m)\big),$$
> since both the bin-width search and the coarsened adjacency construction are repeated for every $r \in R$.
>
> In contrast, AH-UGC performs LSH projections only once in
> $\mathcal{O}(NLd + m),$
> then sorts the hash scores once in
> $\mathcal{O}(N \log N),$
> and finally generates all $|R|$ coarsened graphs by a single pass over the ordered list in
> $\mathcal{O}(N(1-r))$ where $N(1-r)$ is the number of node reduced to get the smallest coarsened graph.
> The total cost is therefore
> $$T_{AH-UGC, Adaptive} = \mathcal{O}\big(NLd + m + N\log N + N(1-r)\big).$$
>
> Thus, for a single coarsening level, AH-UGC and UGC are both near-linear, but for adaptive multi-resolution coarsening, UGC pays the expensive $\mathcal{O}(NLd + m)$ cost $(1+K)|R|$ times, whereas AH-UGC pays it once and then adds only $\mathcal{O}(N\log N + N(1-r))$, making AH-UGC asymptotically cheaper in the adaptive setting.
>
>
>  **3)** Is *l* a hyperparameter? If so, how is it determined?
>
>  **Ans)** Yes *l* is a hyperparameter in our method. In practice, we treat *l* as the number of random projections / hash functions and select it empirically. We observed that performance is quite stable beyond a certain threshold, and *l=1000* consistently provided a good trade-off between accuracy and runtime across all datasets.

---

> ### Author Response · Authors · 2025-11-20
>
> We thank you for acknowledging the potential and methodological contribution of our work. We appreciate your constructive feedback on the writing and have carefully revised the manuscript accordingly, and we look forward to your assessment of these changes in light of this rebuttal.

---

> ### Author Response · Authors · 2025-11-24
>
> Dear Reviewer,
>
> We appreciate your time and feedback on our submission. As the discussion deadline is approaching, we wanted to briefly check in. Your earlier comment, *“I like the methodology of this paper and I think that if the paper can improve their writing, the technical contribution of the paper should be a borderline accept,”* was very encouraging. We have carefully revised the paper and rebuttal to address your concerns, and we would be very grateful if you could let us know whether these changes resolve your questions and, if so, we request you to kindly consider updating your rating.
>
> Thank you again for your time and consideration.

---

> ### Author Response · Authors · 2025-11-28
> **Appeal to Reviewer,**
>
> Dear Reviewer,
>
> Just a quick reminder as the discussion phase is closing soon. We truly appreciated your encouraging feedback and have addressed all your points in detail during the rebuttal. **If you’ve had a chance to review the updates, we’d be very grateful if you could share your final thoughts or consider updating your rating.**
>
> Best regards,
> Authors

---

### Meta-Review · Area_Chair_4roN · 2026-01-06

**Summary:**

Multiple reviewers found the extension over UGC incremental, questioning the depth of theoretical contribution. Writing and notation clarity is poor, highlighted inconsistent notation, unclear goals, and poor organization, which hindered understanding. Claims about streaming applicability and comparisons with recent methods were initially unsupported. Experimentation concerns included missing baselines, incomplete analysis, and unclear experimental choices.

**Reviewer Concerns:**

Addressed concnerns:
Notation and writing clarity were improved via a symbol table, revised text, and structural adjustments. Streaming experiments, additional baselines,ablation studies were added.

Remaining concerns:
The core technical advance was still viewed by multiple reviewers as incremental. Its necessity and benefits remained weakly motivated, with limited theoretical guarantees. Doubts persisted about the framework's universality, especially regarding transformer-based graph models and diverse graph types. While improvements were made, the work was not seen as sufficiently groundbreaking for acceptance, particularly given the strong initial reservations.

**Reviewer Scores:**

Reviewer rpLH (4→ 4): Likely unchanged given unresolved novelty concern.

Reviewer jTzn (6→ 6): Likely unchanged; already positive.

Reviewer M9ms (4→ 4): Likely unchanged due to insufficient comparision.

Reviewer EQiL (4→ 4): Likely unchanged due to persistent issues in novelty.

---

### Decision · Program_Chairs · 2026-01-26

Reject